# VIREL: A Variational Inference Framework for Reinforcement Learning

**Matthew Fellows**[*]   **Anuj Mahajan**[*]   **Tim G. J. Rudner**   **Shimon Whiteson**
Department of Computer Science
University of Oxford

## Abstract

Applying probabilistic models to reinforcement learning (RL) enables the uses of powerful optimisation tools such as variational inference in RL. However, existing inference frameworks and their algorithms pose significant challenges for learning optimal policies, for example, the lack of mode capturing behaviour in pseudo-likelihood methods, difficulties learning deterministic policies in maximum entropy RL based approaches, and a lack of analysis when function approximators are used. We propose VIREL, a theoretically grounded inference framework for RL that utilises a parametrised action-value function to summarise future dynamics of the underlying MDP, generalising existing approaches. VIREL also benefits from a mode-seeking form of KL divergence, the ability to learn deterministic optimal polices naturally from inference, and the ability to optimise value functions and policies in separate, iterative steps. Applying variational expectation-maximisation to VIREL, we show that the actor-critic algorithm can be reduced to expectation-maximisation, with policy improvement equivalent to an E-step and policy evaluation to an M-step. We derive a family of actor-critic methods from VIREL, including a scheme for adaptive exploration and demonstrate that our algorithms outperform state-of-the-art methods based on soft value functions in several domains.

## 1 Introduction

Efforts to combine reinforcement learning (RL) and probabilistic inference have a long history, spanning diverse fields such as control, robotics, and RL [64, 62, 46, 47, 27, 74, 75, 73, 36]. Formalising RL as probabilistic inference enables the application of many approximate inference tools to reinforcement learning, extending models in flexible and powerful ways [35]. However, existing methods at the intersection of RL and inference suffer from several deficiencies. Methods that derive from the pseudo-likelihood inference framework [12, 64, 46, 26, 44, 1] and use expectation-maximisation (EM) favour risk-seeking policies [34], which can be suboptimal. Yet another approach, the MERL inference framework [35] (which we refer to as MERLIN), derives from maximum entropy reinforcement learning (MERL) [33, 74, 75, 73]. While MERLIN does not suffer from the issues of the pseudo-likelihood inference framework, it presents different practical difficulties. These methods do not naturally learn deterministic optimal policies and constraining the variational policies to be deterministic renders inference intractable [47]. As we show by way of counterexample in Section 2.2, an optimal policy under the reinforcement learning objective is not guaranteed from the optimal MERL objective. Moreover, these methods rely on soft value functions which are sensitive to a pre-defined temperature hyperparameter.

Additionally, no existing framework formally accounts for replacing exact value functions with function approximators in the objective; learning function approximators is carried out independently of the inference problem and no analysis of convergence is given for the corresponding algorithms.

---

[*]Equal Contribution. Correspondence to `matthew.fellows@cs.ox.ac.uk` and `anuj.mahajan@cs.ox.ac.uk`.

This paper addresses these deficiencies. We introduce VIREL, an inference framework that translates the problem of finding an optimal policy into an inference problem. Given this framework, we demonstrate that applying EM induces a family of actor-critic algorithms, where the E-step corresponds exactly to policy improvement and the M-step exactly to policy evaluation. Using a variational EM algorithm, we derive analytic updates for both the model and variational policy parameters, giving a unified approach to learning parametrised value functions and optimal policies.

We extensively evaluate two algorithms derived from our framework against DDPG [38] and an existing state-of-the-art actor-critic algorithm, soft actor-critic (SAC) [25], on a variety of OpenAI gym domains [9]. While our algorithms perform similarly to SAC and DDPG on simple low dimensional tasks, they outperform them substantially on complex, high dimensional tasks.

The main contributions of this work are: 1) an exact reduction of entropy regularised RL to probabilistic inference using value function estimators; 2) the introduction of a theoretically justified general framework for developing inference-style algorithms for RL that incorporate the uncertainty in the optimality of the action-value function, $\hat{Q}_\omega(h)$, to drive exploration, but that can also learn optimal deterministic policies; and 3) a family of practical algorithms arising from our framework that adaptively balances exploration-driving entropy with the RL objective and outperforms the current state-of-the-art SAC, reconciling existing advanced actor critic methods like A3C [43], MPO [1] and EPG [10] into a broader theoretical approach.

## 2    Background

We assume familiarity with probabilistic inference [30] and provide a review in Appendix A.

### 2.1    Reinforcement Learning

Formally, an RL problem is modelled as a Markov decision process (MDP) defined by the tuple $\langle \mathcal{S}, \mathcal{A}, r, p, p_0, \gamma \rangle$ [54, 59], where $\mathcal{S}$ is the set of states and $\mathcal{A} \subseteq \mathbb{R}^n$ the set of available actions. An agent in state $s \in \mathcal{S}$ chooses an action $a \in \mathcal{A}$ according to the policy $a \sim \pi(\cdot|s)$, forming a state-action pair $h \in \mathcal{H}$, $h := \langle s, a \rangle$. This pair induces a scalar reward according to the reward function $r_t := r(h_t) \in \mathbb{R}$ and the agent transitions to a new state $s' \sim p(\cdot|h)$. The initial state distribution for the agent is given by $s_0 \sim p_0$. We denote a sampled state-action pair at timestep $t$ as $h_t := \langle s_t, a_t \rangle$. As the agent interacts with the environment using $\pi$, it gathers a trajectory $\tau = (h_0, r_0, h_1, r_1, ...)$. The value function is the expected, discounted reward for a trajectory, starting in state $s$. The action-value function or $Q$-function is the expected, discounted reward for each trajectory, starting in $h$, $Q^\pi(h) := \mathbb{E}_{\tau \sim p^\pi(\tau|h)} \left[ \sum_{t=0}^\infty \gamma^t r_t \right]$, where $p^\pi(\tau|h) := p(s_1|h_0 = h) \prod_{t'=1}^\infty p(s_{t'+1}|h_{t'}) \pi(a_t|s_t)$. Any $Q$-function satisfies a Bellman equation $\mathcal{T}^\pi Q^\pi(\cdot) = Q^\pi(\cdot)$ where $\mathcal{T}^\pi \cdot := r(h) + \gamma \mathbb{E}_{h' \sim p(s'|h)\pi(a'|s')} [\cdot]$ is the Bellman operator. We consider infinite horizon problems with a discount factor $\gamma \in [0, 1)$. The agent seeks an optimal policy $\pi^* \in \arg\max_\pi J^\pi$, where

$$J^\pi = \mathbb{E}_{h \sim p_0(s)\pi(a|s)} \left[ Q^\pi(h) \right]. \tag{1}$$

We denote optimal $Q$-functions as $Q^*(\cdot) := Q^{\pi^*}(\cdot)$ and the set of optimal policies $\Pi^* := \arg\max_\pi J^\pi$. The optimal Bellman operator is $\mathcal{T}^* \cdot := r(h) + \gamma \mathbb{E}_{h' \sim p(s'|h)} [\max_{a'}(\cdot)]$.

### 2.2    Maximum Entropy RL

The MERL objective supplements each reward in the RL objective with an entropy term [61, 74, 75, 73], $J_{\text{merl}}^\pi := \mathbb{E}_{\tau \sim p(\tau)} \left[ \sum_{t=0}^{T-1} (r_t - c\log(\pi(a_t|s_t))) \right]$. The standard RL, undiscounted objective is recovered for $c \to 0$ and we assume $c = 1$ without loss of generality. The MERL objective is often used to motivate the MERL inference framework (which we call MERLIN) [34], mapping the problem of finding the optimal policy, $\pi_{\text{merl}}^*(a|s) = \arg\max_\pi J_{\text{merl}}^\pi$, to an equivalent inference problem. A full exposition of this framework is given by Levine [35] and we discuss the graphical model of MERLIN in comparison to VIREL in Section 3.3. The inference problem is often solved using a message passing algorithm, where the log backward messages are called soft value functions due to their similarity to classic (hard) value functions [63, 48, 25, 24, 35]. The soft $Q$-function is defined as $Q_{\text{soft}}^\pi(h) := \mathbb{E}_{\tau \sim q^\pi(\tau|h)} \left[ r_0 + \sum_{t=1}^{T-1} (r_t - \log \pi(a_t|s_t)) \right]$, where $q^\pi(\tau|h) := p(s_0|h) \prod_{t=0}^{T-1} p(s_{t+1}|h_t)\pi(a_t|s_t)$.

The corresponding soft Bellman operator is $\mathcal{T}_{\text{soft}}^{\pi} \cdot := r(h) + \mathbb{E}_{h' \sim p(s'|h)\pi(a'|s')}[\cdot - \log \pi(a'|s')]$. Several algorithms have been developed that mirror existing RL algorithms using soft Bellman equations, including maximum entropy policy gradients [35], soft $Q$-learning [24], and soft actor-critic (SAC) [25]. MERL is also compatible with methods that use recall traces [21].

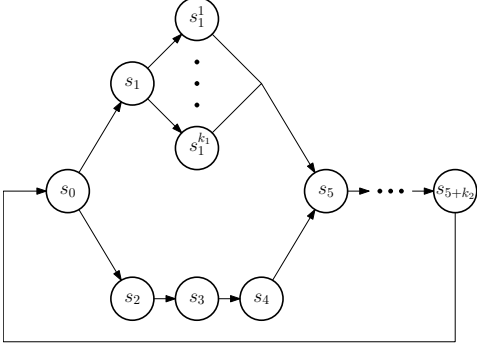

Figure 1: A discrete MDP counterexample for optimal policy under maximum entropy.

We now outline key drawbacks of MERLIN. It is well-understood that optimal policies under regularised Bellman operators are more stochastic than under their equivalent unregularised operators [20]. While this can lead to improved exploration, the optimal policy under these operators will still be stochastic, meaning optimal deterministic policies are not learnt naturally. This leads to two difficulties: 1) a deterministic policy can be constructed by taking the action $a^* = \arg\max_a \pi_{\text{merl}}^*(a|s)$, corresponding to the maximum a posteriori (MAP) policy, however, in continuous domains, finding the MAP policy requires optimising the $Q$-function approximator for actions, which is often a deep neural network. A common approximation is to use the mean of a variational policy instead; 2) even if we obtain a good approximation, as we show below by way of counterexample, the deterministic MAP policy is not guaranteed to be the optimal policy under $J^{\pi}$. Constraining the variational policies to the set of Dirac-delta distributions does not solve this problem either, since it renders the inference procedure intractable [47, 48].

Next, we demonstrate that the optimal policy under $J^{\pi}$ cannot always be recovered from the MAP policy under $J_{\text{merl}}^{\pi}$. Consider the discrete state MDP as shown in Fig. 1, with action set $\mathcal{A} = \{a_1, a_2, a_1^1, \cdots a_1^{k_1}\}$ and state set $\mathcal{S} = \{s_0, s_1, s_2, s_3, s_4, s_1^1 \cdots s_1^{k_1}, s_5, \cdots s_{5+k_2}\}$. All state transitions are deterministic, with $p(s_1|s_0, a_1) = p(s_1|s_0, a_2) = p(s_1^i|s_1, a_1^i) = 1$. All other state transitions are deterministic and independent of action taken, that is, $p(s_j|\cdot, s_{j-1}) = 1 \ \forall \ j > 2$ and $p(s_5|\cdot, s_1^i) = 1$. The reward function is $r(s_0, a_2) = 1$ and zero otherwise. Clearly the optimal policy under $J^{\pi}$ has $\pi^*(a_2|s_0) = 1$. Define a maximum entropy reinforcement learning policy as $\pi_{\text{merl}}$ with $\pi_{\text{merl}}(a_1|s_0) = p_1$, $\pi_{\text{merl}}(a_2|s_0) = (1 - p_1)$ and $\pi_{\text{merl}}(a_1^i|s_1) = p_1^i$. For $\pi_{\text{merl}}$ and $k_2 >> 5$, we can evaluate $J_{\text{merl}}^{\pi}$ for any scaling constant $c$ and discount factor $\gamma$ as:

$$J_{\text{merl}}^{\pi} = (1 - p_1)(1 - c\log(1 - p_1)) - p_1\left(c\log p_1 + \gamma c\sum_{i=1}^{k} p_1^i \log p_1^i\right). \tag{2}$$

We now find the optimal MERL policy. Note that $p_1^i = \frac{1}{k}$ maximises the final term in Eq. (2). Substituting for $p_1^i = \frac{1}{k_1}$, then taking derivatives of Eq. (2) with respect to $p_1$, and setting to zero, we find $p_1^* = \pi_{\text{merl}}^*(a_1|s_0)$ as:

$$1 - c\log(1 - p_1^*) = \gamma c\log(k_1) - c\log p_1^*,$$

$$\implies p_1^* = \frac{1}{k_1^{-\gamma}\exp\left(\frac{1}{c}\right) + 1},$$

hence, for any $k_1^{-\gamma}\exp\left(\frac{1}{c}\right) < 1$, we have $p_1^* > \frac{1}{2}$ and so $\pi^*$ cannot be recovered from $\pi_{\text{merl}}^*$, even using the mode action $a_1 = \arg\max_a \pi_{\text{merl}}^*(a|s_0)$. The degree to which the MAP policy varies from the optimal unregularised policy depends on both the value of $c$ and $k_1$, the later controlling the number of states with sub-optimal reward. Our counterexample illustrates that when there are large regions of the state-space with sub-optimal reward, the temperature must be comparatively small to compensate, hence algorithms derived from MERLIN become very sensitive to temperature. As we discuss in Section 3.3, this problem stems from the fact that MERL policies optimise for expected reward and long-term expected entropy. While initially beneficial for exploration, this can lead to sub-optimal polices being learnt in complex domains as there is often too little a priori knowledge about the MDP to make it possible to choose an appropriate value or schedule for $c$.

Finally, a minor issue with MERLIN is that many existing models are defined for finite-horizon problems [35, 48]. While it is possible to discount and extend MERLIN to infinite-horizon problems, doing so is often nontrivial and can alter the objective [60, 25].

## 2.3 Pseudo-Likelihood Methods

A related but distinct approach is to apply Jensen's inequality directly to the RL objective $J^\pi$. Firstly, we rewrite Eq. (1) as an expectation over $\tau$ to obtain $J = \mathbb{E}_{h \sim p_0(s)\pi(a|s)} [Q^\pi(h)] = \mathbb{E}_{\tau \sim p(\tau)} [R(\tau)]$, where $R(\tau) = \sum_{t=0}^{T-1} \gamma^t r_t$ and $p(\tau) = p_0(s_0)\pi(a_0|s_o) \prod_{t=0}^{T-1} p(h_{t+1}|h_t)$. We then treat $p(R, \tau) = R(\tau)p(\tau)$ as a joint distribution, and if rewards are positive and bounded, Jensen's inequality can be applied, enabling the derivation of an evidence lower bound (ELBO). Inference algorithms such as EM can then be employed to find a policy that optimises the pseudo-likelihood objective [12, 64, 46, 26, 44, 1]. Pseudo-likelihood methods can also be extended to a model-based setting by defining a prior over the environment's transition dynamics. Furmston & Barber [19] demonstrate that the posterior over all possible environment models can be integrated over to obtain an optimal policy in a Bayesian setting.

Many pseudo-likelihood methods minimise $\mathrm{KL}(p_\mathcal{O} \parallel p_\pi)$, where $p_\pi$ is the policy to be learnt and $p_\mathcal{O}$ is a target distribution monotonically related to reward [35]. Classical RL methods minimise $\mathrm{KL}(p_\pi \parallel p_\mathcal{O})$. The latter encourages learning a mode of the target distribution, while the former encourages matching the moments of the target distribution. If the optimal policy can be represented accurately in the class of policy distributions, optimisation converges to a global optimum and the problem is fully observable, the optimal policy is the same in both cases. Otherwise, the pseudo-likelihood objective reduces the influence of large negative rewards, encouraging risk-seeking policies.

## 3 VIREL

Before describing our framework, we state some relevant assumptions.

**Definition 1** (Unique Maximum and Locally Smooth Function). *Let $f : \mathcal{X} \to \mathcal{Y}$ be a function with a unique maximum $f(x^*) = \sup_x f$ where the domain $\mathcal{X}$ is a compact set and range $\mathcal{Y}$ is bounded. Let $f$ be locally $\mathbb{C}^2$ smooth about $x^*$, i.e., $\exists \Delta > 0$ s.t. $f(x) \in \mathbb{C}^2 \ \forall \ x \in \{x | \|x - x^*\| < \Delta \}$.*

**Assumption 1.** *The optimal action-value function for the reinforcement learning problem is finite and strictly positive, i.e., $0 < Q^*(h) < \infty \ \forall \ h \in \mathcal{H}$.*

Any MDP for which rewards are lower bounded and finite, that is, $R \subset [r_{\min}, \infty)$, satisfies Assumption 1. To see this, we can construct a new MDP by adding $r_{\min}$ to the reward function, ensuring that all rewards are positive and hence the optimal action-value function for the reinforcement learning problem is finite and strictly positive. This does not affect the optimal solution. Now we introduce a function approximator $\hat{Q}_\omega(h) \approx Q^\pi(h)$ parametrised by $\omega \in \Omega$.

**Assumption 2** (Exact Representability Under Optimisation). *Our function approximator can represent the optimal Q-function, i.e., $\exists \ \omega^* \in \Omega$ s.t. $Q^*(\cdot) = \hat{Q}_{\omega^*}(\cdot)$.*

In Appendix F.1, we extend the work of Bhatnagar et al. [6] to continuous domains, demonstrating that Assumption 2 can be neglected if projected Bellman operators are used.

**Assumption 3** (Local Smoothness of Q-functions ). *For $\omega^*$ parametrising $Q^*(h)$ in Assumption 2, $Q_{\omega^*}(h)$ has a unique maximum and is locally smooth under Definition 1 for actions in any state.*

This assumption is formally required for the strict convergence of a Boltzmann to a Dirac-delta distribution and, as we discuss in Appendix F.4, is of more mathematical than practical concern.

### 3.1 Objective Specification

We now define an objective that we motivate by satisfying three desiderata: ① In the limit of maximising our objective, a deterministic optimal policy can be recovered and the optimal Bellman equation is satisfied by our function approximator; ② when our objective is not maximised, stochastic policies can be recovered that encourage effective exploration of the state-action space; and ③ our objective permits the application of powerful and tractable optimisation algorithms from variational inference that optimise the risk-neutral form of KL divergence, $\mathrm{KL}(p_\pi \parallel p_\mathcal{O})$, introduced in Section 2.3.

Firstly, we define the residual error $\varepsilon_\omega := \frac{c}{p} \|\mathcal{T}_\omega \hat{Q}_\omega(h) - \hat{Q}_\omega(h)\|_p^p$ where $\mathcal{T}_\omega = \mathcal{T}^{\pi_\omega} \cdot := r(h) + \gamma \mathbb{E}_{h' \sim p(s'|h)\pi_\omega(a'|s')} [\cdot]$ is the Bellman operator for the Boltzmann policy with temperature $\varepsilon_\omega$:

$$\pi_\omega(a|s) := \frac{\exp\left(\frac{\hat{Q}_\omega(h)}{\varepsilon_\omega}\right)}{\int_\mathcal{A} \exp\left(\frac{\hat{Q}_\omega(h)}{\varepsilon_\omega}\right) da}. \tag{3}$$

We assume $p = 2$ and $c = \frac{1}{|\mathcal{H}|}$ without loss of generality. Our main result in Theorem 2 proves that finding a $\omega^*$ that reduces the residual error to zero, i.e., $\varepsilon_{\omega^*} = 0$, is a sufficient condition for learning an optimal $Q$-function $\hat{Q}_{\omega^*}(h) = Q^*(h)$. Additionally, the Boltzmann distribution $\pi_\omega(a|s)$ tends towards a Dirac-delta distribution $\pi_\omega(a|s) = \delta(a = \arg\max'_a \hat{Q}_{\omega^*}(a', s))$ whenever $\varepsilon_\omega \to 0$ (see Theorem 1), which is an optimal policy. The simple objective $\arg\min(\mathcal{L}(\omega)) \coloneqq \arg\min(\varepsilon_\omega)$ therefore satisfies ①. Moreover, when our objective is not minimised, we have $\varepsilon_\omega > 0$ and from Eq. (3) we see that $\pi_\omega(a|s)$ is non-deterministic *for all non-optimal* $\omega$. $\mathcal{L}(\omega)$ therefore satisfies ② as any agent following $\pi_\omega(a|s)$ will continue exploring until the RL problem is solved. To generalise our framework, we extend $\mathcal{T}_\omega \cdot$ to any operator from the set of target operators $\mathcal{T}_\omega \cdot \in \mathbb{T}$:

**Definition 2** (Target Operator Set). *Define $\mathbb{T}$ to be the set of target operators such that an optimal Bellman operator for $\hat{Q}_\omega(h)$ is recovered when the Boltzmann policy in Eq.* (3) *is greedy with respect to $\hat{Q}_\omega(h)$, i.e., $\mathbb{T} \coloneqq \{\mathcal{T}_\omega \cdot \,|\, \lim_{\varepsilon_\omega \to 0} \pi_\omega(a|s) \implies \mathcal{T}_\omega \hat{Q}_\omega(h) = \mathcal{T}^* \hat{Q}_\omega(h)\}$.*

As an illustration, we prove in Appendix C that the Bellman operator $\mathcal{T}^{\pi_\omega} \cdot$ introduced above is a member of $\mathbb{T}$ and can be approximated by several well-known RL targets. We also discuss how $\mathcal{T}^{\pi_\omega} \cdot$ induces a constraint on $\Omega$ due to its recursive definition. As we show in Section 3.2, there exists an $\omega$ in the constrained domain that maximises the RL objective under these conditions, so an optimal solution is always feasible. Moreover, we provide an analysis in Appendix F.5 to establish that such a policy is an attractive fixed point for our algorithmic updates, even when we ignore this constraint. Off-policy operators will not constrain $\Omega$: by definition, the optimal Bellman operator $\mathcal{T}^* \cdot$ is a member of $\mathbb{T}$ and does not constrain $\Omega$; similarly, we derive an off-policy operator based on a Boltzmann distribution with a diminishing temperature in Appendix F.2 that is a member of $\mathbb{T}$. Observe that soft Bellman operators are not members of $\mathbb{T}$ as the optimal policy under $J^\pi_{\text{merl}}$ is not deterministic, hence algorithms such as SAC cannot be derived from the VIREL framework.

One problem remains: calculating the normalisation constant to sample directly from the Boltzmann distribution in Eq. (3) is intractable for many MDPs and function approximators. As such, we look to variational inference to learn an approximate *variational policy* $\pi_\theta(a|s) \approx \pi_\omega(a|s)$, parametrised by $\theta \in \Theta$ with finite variance and the same support as $\pi_\omega(a|s)$. This suggests optimising a new objective that penalises $\pi_\theta(a|s)$ when $\pi_\theta(a|s) \neq \pi_\omega(a|s)$ but still has a global maximum at $\varepsilon_\omega = 0$. A tractable objective that meets these requirements is the evidence lower bound (ELBO) on the unnormalised potential of the Boltzmann distribution, defined as $\{\omega^*, \theta^*\} \in \arg\max_{\omega,\theta} \mathcal{L}(\omega, \theta)$,

$$\mathcal{L}(\omega, \theta) \coloneqq \mathbb{E}_{s \sim d(s)} \left[ \mathbb{E}_{a \sim \pi_\theta(a|s)} \left[ \frac{\hat{Q}_\omega(h)}{\varepsilon_\omega} \right] + \mathscr{H}(\pi_\theta(a|s)) \right], \tag{4}$$

where $q_\theta(h) \coloneqq d(s)\pi_\theta(a|s)$ is a variational distribution, $\mathscr{H}(\cdot)$ denotes the differential entropy of a distribution, and $d(s)$ is any arbitrary sampling distribution with support over $\mathcal{S}$. From Eq. (4), maximising our objective with respect to $\omega$ is achieved when $\varepsilon_\omega \to 0$ and hence $\mathcal{L}(\omega, \theta)$ satisfies ① and ②. As we show in Lemma 1, $\mathscr{H}(\cdot)$ in Eq. (4) causes $\mathcal{L}(\omega, \theta) \to -\infty$ whenever $\pi_\theta(a|s)$ is a Dirac-delta distribution for all $\varepsilon_\omega > 0$. This means our objective heavily penalises premature convergence of our variational policy to greedy Dirac-delta policies except under optimality. We discuss a probabilistic interpretation of our framework in Appendix B, where it can be shown that $\pi_\omega(a|s)$ characterises our model's uncertainty in the optimality of $\hat{Q}_\omega(h)$.

We now motivate $\mathcal{L}(\omega, \theta)$ from an inference perspective: In Appendix D.1, we write $\mathcal{L}(\omega, \theta)$ in terms of the log-normalisation constant of the Boltzmann distribution and the KL divergence between the action-state normalised Boltzmann distribution, $p_\omega(h)$, and the variational distribution, $q_\theta(h)$:

$$\mathcal{L}(\omega, \theta) = \ell(\omega) - \text{KL}(q_\theta(h) \,\|\, p_\omega(h)) - \mathscr{H}(d(s)), \tag{5}$$

$$\text{where} \quad \ell(\omega) \coloneqq \log \int_\mathcal{H} \exp\left(\frac{\hat{Q}_\omega(h)}{\varepsilon_\omega}\right) dh, \quad p_\omega(h) \coloneqq \frac{\exp\left(\frac{\hat{Q}_\omega(h)}{\varepsilon_\omega}\right)}{\int_\mathcal{H} \exp\left(\frac{\hat{Q}_\omega(h)}{\varepsilon_\omega}\right) dh}.$$

As the KL divergence in Eq. (5) is always positive and the final entropy term has no dependence on $\omega$ or $\theta$, maximising our objective for $\theta$ always reduces the KL divergence between $\pi_\omega(a|s)$ and $\pi_\theta(a|s)$ for any $\varepsilon_\omega > 0$, with $\pi_\theta(a|s) = \pi_\omega(a|s)$ achieved under exact representability (see Theorem 3). This yields a tractable way to estimate $\pi_\omega(a|s)$ at any point during our optimisation procedure by maximising $\mathcal{L}(\omega, \theta)$ for $\theta$. From Eq. (5), we see that our objective satisfies ③, as we minimise the

mode-seeking direction of KL divergence, $\text{KL}(q_\theta(h) \| p_\omega(h))$, and our objective is an ELBO, which is the starting point for inference algorithms [30, 4, 17]. When the RL problem is solved and $\varepsilon_\omega = 0$, our objective tends towards infinity for *any* variational distribution that is non-deterministic (see Lemma 1). This is of little consequence, however, as whenever $\varepsilon_\omega = 0$, our approximator is the optimal value function, $\hat{Q}_{\omega^*}(h) = Q^*(h)$ (Theorem 2), and hence, $\pi^*(a|s)$ can be inferred exactly by finding $\max_{a'} \hat{Q}_{\omega^*}(a', s)$ or by using the policy gradient $\nabla_\theta \mathbb{E}_{d(s)\pi_\theta(a|s)} \left[ \hat{Q}_{\omega^*}(h) \right]$ (see Section 4.2).

## 3.2 Theoretical Results

We now formalise the intuition behind ①-③. Theorem 1 establishes the emergence of a Dirac-delta distribution in the limit of $\varepsilon_\omega \to 0$. To the authors' knowledge, this is the first rigorous proof of this result. Theorem 2 shows that finding an optimal policy that maximises the RL objective in Eq. (1) reduces to finding the Boltzmann distribution associated with the parameters $\omega^* \in \arg\max_\omega \mathcal{L}(\omega, \theta)$. The existence of such a distribution is a sufficient condition for the policy to be optimal. Theorem 3 shows that whenever $\varepsilon_\omega > 0$, maximising our objective for $\theta$ always reduces the KL divergence between $\pi_\omega(a|s)$ and $\pi_\theta(a|s)$, providing a tractable method to infer the current Boltzmann policy.

**Theorem 1** (Convergence of Boltzmann Distribution to Dirac Delta). *Let $p_\varepsilon : \mathcal{X} \to [0, 1]$ be a Boltzmann distribution with temperature $\varepsilon \in \mathbb{R}_{\geq 0}$, $p_\varepsilon(x) = \frac{\exp\left(\frac{f(x)}{\varepsilon}\right)}{\int_\mathcal{X} \exp\left(\frac{f(x)}{\varepsilon}\right)dx}$, where $f : \mathcal{X} \to \mathcal{Y}$ is a function that satisfies Definition 1. In the limit $\varepsilon \to 0$, $p_\varepsilon(x) \to \delta(x = \sup_{x'} f(x'))$.*
*Proof.* See Appendix D.2 □

**Lemma 1** (Lower and Upper limits of $\mathcal{L}(\omega, \theta)$). *i) For any $\varepsilon_\omega > 0$ and $\pi_\theta(a|s) = \delta(a^*)$, we have $\mathcal{L}(\omega, \theta) = -\infty$. ii) For $\hat{Q}_\omega(h) > 0$ and any non-deterministic $\pi_\theta(a|s)$, $\lim_{\varepsilon_\omega \to 0} \mathcal{L}(\omega, \theta) = \infty$.*
*Proof.* See Appendix D.3. □

**Theorem 2** (Optimal Boltzmann Distributions as Optimal Policies). *For $\omega^*$ that maximises $\mathcal{L}(\omega, \theta)$ defined in Eq. (4), the corresponding Boltzmann policy induced must be optimal, i.e., $\{\omega^*, \theta^*\} \in \arg\max_{\omega,\theta} \mathcal{L}(\omega, \theta) \implies \pi_{\omega^*}(a|s) \in \Pi^*$.*
*Proof.* See Appendix D.3. □

**Theorem 3** (Maximising the ELBO for $\theta$). *For any $\varepsilon_\omega > 0$, $\max_\theta \mathcal{L}(\omega, \theta) = \mathbb{E}_{d(s)} \left[ \min_\theta \text{KL}(\pi_\theta(a|s) \| \pi_\omega(a|s)) \right]$ with $\pi_\omega(a|s) = \pi_\theta(a|s)$ under exact representability.*
*Proof.* See Appendix D.4. □

## 3.3 Comparing VIREL and MERLIN Frameworks

To compare MERLIN and VIREL, we consider the probabilistic interpretation of the two models discussed in Appendix B; introducing a binary variable $\mathcal{O} \in \{0, 1\}$ defines a graphical model for our inference problem whenever $\varepsilon_\omega > 0$. Comparing the graphs in Fig. 2, observe that MERLIN models exponential *cumulative* rewards over entire trajectories. By contrast, VIREL's variational policy models a single step and a function approximator is used to model future *expected* rewards.

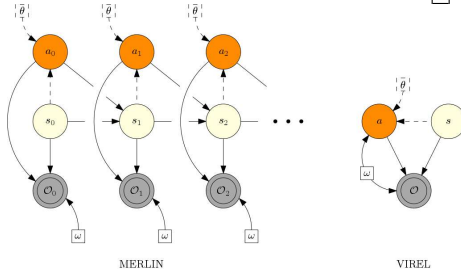

Figure 2: Graphical models for MERLIN and VIREL (variational approximations are dashed).

The resulting KL divergence minimisation for MERLIN is therefore much more sensitive to the value of temperature, as this affects how much future entropy influences the variational policy. For VIREL, temperature is defined by the model, and updates to the variational policy will not be as sensitive to errors in its value or linear scaling as its influence only extends to a single interaction. We hypothesise that VIREL may afford advantages in higher dimensional domains where there is greater chance of encountering large regions of state-action space with sub-optimal reward; like our counterexample from Section 2, $c$ must be comparatively small to balance the influence of entropy in these regions to prevent MERLIN algorithms from learning sub-optimal policies.

Theorem 1 demonstrates that, unlike in MERLIN, VIREL naturally learns optimal deterministic policies directly from the optimisation procedure while still maintaining the benefits of stochastic policies in training. While Boltzmann policies with fixed temperatures have been proposed before [49], as we discuss in Appendix B, the adaptive temperature $\varepsilon_\omega$ in VIREL's Boltzmann policy has a unique interpretation, characterising the model's uncertainty in the optimality of $\hat{Q}_\omega(h)$; both $\pi_\omega(a|s)$ and its variational approximation $\pi_\theta(a|s)$ have an adaptive variance that reduces as $\hat{Q}_\omega(h) \to Q^*(h)$, allowing us to benefit from uncertainty-driven exploration when sampling under $\pi_\theta(a|s)$.

# 4 Actor-Critic and EM

We now apply the expectation-maximisation (EM) algorithm [13, 23] to optimise our objective $\mathcal{L}(\omega, \theta)$. (See Appendix A for an exposition of this algorithm.) In keeping with RL nomenclature, we refer to $\hat{Q}_\omega(h)$ as the *critic* and $\pi_\theta(a|s)$ as the *actor*. We establish that the expectation (E-) step is equivalent to carrying out policy improvement and the maximisation (M-)step to policy evaluation. This formulation reverses the situation in most pseudo-likelihood methods, where the E-step is related to policy evaluation and the M-step is related to policy improvement, and is a direct result of optimising the forward KL divergence, $\text{KL}(q_\theta(h) \parallel p_\omega(h|\mathcal{O}))$, as opposed to the reverse KL divergence used in pseudo-likelihood methods. As discussed in Section 2.3, this mode-seeking objective prevents the algorithm from learning risk-seeking policies. We now introduce an extension to Assumption 2 that is sufficient to guarantee convergence.

**Assumption 4** (Universal Variational Representability). *Every Boltzmann policy can be represented as $\pi_\theta(a|s)$, i.e., $\forall\, \omega \in \Omega\, \exists\, \theta \in \Theta$ s.t. $\pi_\theta(a|s) = \pi_\omega(a|s)$.*

Assumption 4 is strong but, like in variational inference, our variational policy $\pi_\theta(a|s)$ provides a useful approximation when Assumption 4 does not hold. As we discuss in Appendix F.1, using projected Bellman errors also ensures that our M-step always converges no matter what our current policy is.

## 4.1 Variational Actor-Critic

In the E-step, we keep the parameters of our critic $\omega_k$ constant while updating the actor's parameters by maximising the ELBO with respect to $\theta$: $\theta_{k+1} \leftarrow \arg\max_\theta \mathcal{L}(\omega_k, \theta)$. Using gradient ascent with step size $\alpha_{\text{actor}}$, we optimise $\varepsilon_{\omega_k} \mathcal{L}(\omega_k, \theta)$ instead, which prevents ill-conditioning and does not alter the optimal solution, yielding the update (see Appendix E.1 for full derivation):

**E-Step (Actor):** $\quad \theta_{i+1} \leftarrow \theta_i + \alpha_{\text{actor}} \left( \varepsilon_{\omega_k} \nabla_\theta \mathcal{L}(\omega_k, \theta) \right)|_{\theta=\theta_i},$

$$\varepsilon_{\omega_k} \nabla_\theta \mathcal{L}(\omega_k, \theta) = \mathbb{E}_{s \sim d(s)} \left[ \mathbb{E}_{a \sim \pi_\theta(a|s)} \left[ \hat{Q}_{\omega_k}(h) \nabla_\theta \log \pi_\theta(a|s) \right] + \varepsilon_{\omega_k} \nabla_\theta \mathscr{H}(\pi_\theta(a|s)) \right]. \quad (6)$$

In the M-step, we maximise the ELBO with respect to $\omega$ while holding the parameters $\theta_{k+1}$ constant. Hence expectations are taken with respect to the variational policy found in the E-step: $\omega_{k+1} \leftarrow \arg\max_\omega \mathcal{L}(\omega, \theta_{k+1})$. We use gradient ascent with step size $\alpha_{\text{critic}}(\varepsilon_{\omega_i})^2$ to optimise $\mathcal{L}(\omega, \theta_{k+1})$ to prevent ill-conditioning, yielding (see Appendix E.2 for full derivation):

**M-Step (Critic):** $\quad \omega_{i+1} \leftarrow \omega_i + \alpha_{\text{critic}}(\varepsilon_{\omega_i})^2 \nabla_\omega \mathcal{L}(\omega, \theta_{k+1})|_{\omega=\omega_i},$

$$(\varepsilon_{\omega_i})^2 \nabla_\omega \mathcal{L}(\omega, \theta_{k+1}) = \varepsilon_{\omega_i} \mathbb{E}_{d(s)\pi_{\theta_{k+1}}(a|s)} \left[ \nabla_\omega \hat{Q}_\omega(h) \right] - \mathbb{E}_{d(s)\pi_{\theta_{k+1}}(a|s)} \left[ \hat{Q}_{\omega_i}(h) \right] \nabla_\omega \varepsilon_\omega. \quad (7)$$

## 4.2 Discussion

From an RL perspective, the E-step corresponds to training an actor using a policy gradient method [56] with an adaptive entropy regularisation term [69, 43]. The M-step update corresponds to a policy evaluation step, as we seek to reduce the MSBE in the second term of Eq. (7). We derive $\nabla_\omega \varepsilon_\omega$ exactly in Appendix E.3. Note that this term depends on $(\mathcal{T}_\omega \hat{Q}_\omega(h) - \hat{Q}_\omega(h)) \nabla_\omega \mathcal{T}_\omega \hat{Q}_\omega(h)$, which typically requires evaluating two independent expectations. For convergence guarantees, techniques such as residual gradients [2] or GTD2/TDC [6] need to be employed to obtain an unbiased estimate of this term. If guaranteed convergence is not a priority, dropping gradient terms allows us to use direct methods [55], which are often simpler to implement. We discuss these methods further in Appendix F.3 and provide an analysis in Appendix F.5 demonstrating that the corresponding updates act as a variational approximation to $Q$-learning [68, 42]. A key component of our algorithm is the behaviour when $\varepsilon_{\omega^*} = 0$; under this condition, there is no M-step update (both $\varepsilon_{\omega_k} = 0$ and $\nabla_\omega \varepsilon_\omega = 0$) and $Q_{\omega^*}(h) = Q^*(h)$ (see Theorem 2), so our E-step reduces exactly to a policy gradient step, $\theta_{k+1} \leftarrow \theta_k + \alpha_{\text{actor}} \mathbb{E}_{h \sim d(s)\pi_\theta(a|s)} [Q^*(h) \nabla_\theta \log \pi_\theta(a|s)]$, recovering the optimal policy in the limit of convergence, that is, $\pi_\theta(a|s) \to \pi^*(a|s)$.

From an inference perspective, the E-step improves the parameters of our variational distribution to reduce the gap between the current Boltzmann posterior and the variational policy, $\text{KL}(\pi_\theta(a|s)) \parallel \pi_{\omega_k}(a|s))$ (see Theorem 3). This interpretation makes precise the intuition that how much we can improve our policy is determined by how similar $\hat{Q}_{\omega_k}(h)$ is to $Q^*(h)$, limiting

policy improvement to the complete E-step: $\pi_{\theta_{k+1}}(a|s) = \pi_{\omega_k}(a|s)$. We see that the common greedy policy improvement step, $\pi_{\theta_{k+1}}(a|s) = \delta(a \in \arg\max_{a'}(\hat{Q}_{\omega_k}(a', s)))$ acts as an approximation to the Boltzmann form in Eq. (3), replacing the softmax with a hard maximum.

If Assumption 4 holds and any constraint induced by $\mathcal{T}_\omega\cdot$ does not prevent convergence to a complete E-step, the EM algorithm alternates between two convex optimisation schemes, and is guaranteed to converge to at least a local optimum of $\mathcal{L}(\omega, \theta)$ [71]. In reality, we cannot carry out complete E- and M-steps for complex domains, and our variational distributions are unlikely to satisfy Assumption 4. Under these conditions, we can resort to the empirically successful variational EM algorithm [30], carrying out partial E- and M-steps instead, which we discuss further in Appendix F.3.

### 4.3 Advanced Actor-Critic Methods

A family of actor-critic algorithms follows naturally from our framework: 1) we can use powerful inference techniques such as control variates [22] or variance-reducing baselines by subtracting any function that does not depend on the action [50], e.g., $V(s)$, from the action-value function, as this does not change our objective, 2) we can manipulate Eq. (6) to obtain variance-reducing gradient estimators such as EPG [11], FPG [15], and SVG0 [28], and 3) we can take advantage of $d(s)$ being any general decorrelated distribution by using replay buffers [42] or empirically successful asynchronous methods that combine several agents' individual gradient updates at once [43]. As we discuss in Appendix E.4, the manipulation required to derive the estimators in 2) is not strictly justified in the classic policy gradient theorem [56] and MERL formulation [25].

MPO is a state-of-the-art EM algorithm derived from the pseudo-likelihood objective [1]. In its derivation, policy evaluation does not naturally arise from either of its EM steps and must be carried out separately. In addition, its E step is approximated, giving rise to the the one step KL regularised update. As we demonstrate in Appendix G, under the probabilistic interpretation of our model, including a prior of the form $p_\phi(h) = \mathcal{U}(s)\pi_\phi(a|s)$ in our ELBO and specifying a hyper-prior $p(\omega)$, the MPO objective with an adaptive regularisation constant can be recovered from VIREL:

$$\mathcal{L}^{\text{MPO}}(\omega, \theta, \phi) = \mathbb{E}_{s \sim d(s)} \left[ \mathbb{E}_{a \sim \pi_\theta(a|s)} \left[ \frac{\hat{Q}_\omega(h)}{\varepsilon_\omega} \right] - \text{KL}(\pi_\theta(a|s) \parallel \pi_\phi(a|s)) \right] + \log p(\omega).$$

We also show in Appendix G that applying the (variational) EM algorithm from Section 4 yields the MPO updates with the missing policy evaluation step and without approximation in the E-step.

## 5 Experiments

We evaluate our EM algorithm using the direct method approximation outlined in Appendix F.3 with $\mathcal{T}_\omega$, ignoring constraints on $\Omega$. The aim of our evaluation is threefold: Firstly, as explained in Section 3.1, algorithms using soft value functions cannot be recovered from VIREL. We therefore demonstrate that using hard value functions does not affect performance. Secondly, we provide evidence for our hypothesis stated in Section 3.3 that using soft value functions may harm performance in higher dimensional tasks. Thirdly, we show that even under all practical approximations discussed, the algorithm derived in Section 4 still outperforms advanced actor-critic methods.

We compare our methods to the state-of-the-art SAC[2] and DDPG [38] algorithms on MuJoCo tasks in OpenAI gym [9] and in rllab [14]. We use SAC as a baseline because Haarnoja et al. [25] show that it outperforms PPO [52], Soft $Q$-Learning [24], and TD3 [18]. We compare to DDPG [38] because, like our methods, it can learn deterministic optimal policies. We consider two variants: In the first one, called *virel*, we keep the scale of the entropy term in the gradient update for the variational policy constant $\alpha$; in the second, called *beta*, we use an estimate $\hat{\varepsilon}_\omega$ of $\varepsilon_\omega$ to scale the corresponding term in Eq. (25). We compute $\hat{\varepsilon}_\omega$ using a buffer to draw a fixed number of samples $N_\varepsilon$ for the estimate.

To adjust for the relative magnitude of the first term in Eq. (25) with that of $\varepsilon_\omega$ scaling the second term, we also multiply the estimate $\hat{\varepsilon}_\omega$ by a scalar $\lambda \approx \frac{1-\gamma}{r_{avg}}$, where $r_{avg}$ is the average reward observed; $\lambda^{-1}$ roughly captures the order of magnitude of the first term and allows $\hat{\varepsilon}_\omega$ to balance policy changes

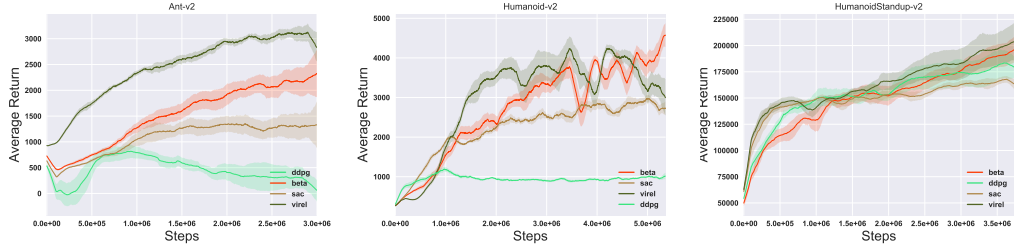

Figure 3: Training curves on continuous control benchmarks gym-Mujoco-v2 : High-dimensional domains.

between exploration and exploitation. We found performance is poor and unstable without $\lambda$. To reduce variance, all algorithms use a value function network $V(\phi)$ as a baseline and a Gaussian policy, which enables the use of the reparametrisation trick. Pseudocode can be found in Appendix H. All experiments use 5 random initialisations and parameter values are given in Appendix I.1.

Fig. 3 gives the training curves for the various algorithms on high-dimensional tasks for on gym-mujoco-v2. In particular, in Humanoid-v2 (action space dimensionality: 17, state space dimensionality: 376) and Ant-v2 (action space dimensionality: 8, state space dimensionality: 111), DDPG fails to learn any reasonable policy. We believe that this is because the Ornstein-Uhlenbeck noise that DDPG uses for exploration is insufficiently adaptive in high dimensions. While SAC performs better, *virel* and *beta* still significantly outperform it. As hypothesised in Section 3.3, we believe that this performance advantage arises because the gap between optimal unregularised policies and optimal variational policies learnt under MERLIN is sensitive to temperature $c$. This effect is exacerbated in high dimensions where there may be large regions of the state-action space with sub-optimal reward. All algorithms learn optimal policies in simple domains, the training curves for which can be found in Fig. 8 in Appendix I.3. Thus, as the state-action dimensionality increases, algorithms derived from VIREL outperform SAC and DDPG.

Fujimoto et al. [18] and van Hasselt et al. [67] note that using the minimum of two randomly initialised action-value functions helps mitigate the positive bias introduced by function approximation in policy gradient methods. Therefore, a variant of SAC uses two soft critics. We compare this variant of SAC to two variants of *virel*: *virel1*, which uses two hard $Q$-functions and *virel2*, which uses one hard and one soft $Q$-function. We scale the rewards so that the means of the $Q$-function estimates in *virel2* are approximately aligned. Fig. 4 shows the training curves on three gym-Mujoco-v1 domains, with additional plots shown in Fig. 7 in Appendix I.2. Again, the results demonstrate that *virel1* and *virel2* perform on par with SAC in simple domains like Half-Cheetah and outperform it in challenging high-dimensional domains like humanoid-gym and -rllab (17 and 21 dimensional action spaces, 376 dimensional state space).

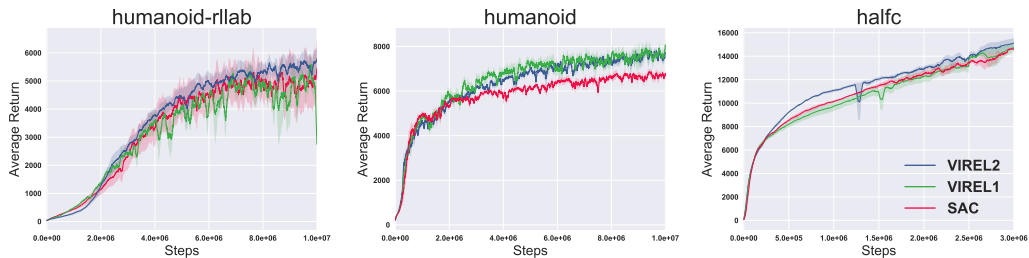

Figure 4: Training curves on continuous control benchmarks gym-Mujoco-v1.

## 6 Conclusion and Future Work

This paper presented VIREL, a novel framework that recasts the reinforcement learning problem as an inference problem using function approximators. We provided strong theoretical justifications for this framework and compared two simple actor-critic algorithms that arise naturally from applying variational EM on the objective. Extensive empirical evaluation shows that our algorithms perform on par with current state-of-the-art methods on simple domains and substantially outperform them on challenging high dimensional domains. As immediate future work, our focus is to find better estimates of $\varepsilon_\omega$ to provide a principled method for uncertainty based exploration; we expect it to help attain sample efficiency in conjunction with various methods like [39, 40]. Another avenue of research would extend our framework to multi-agent settings, in which it can be used to tackle the sub-optimality induced by representational constraints used in MARL algorithms [41].

# 7 Acknowledgements

This project has received funding from the European Research Council (ERC) under the European Unions Horizon 2020 research and innovation programme (grant agreement number 637713). The experiments were made possible by a generous equipment grant from NVIDIA. Matthew Fellows is funded by the EPSRC. Anuj Mahajan is funded by Google DeepMind and the Drapers Scholarship. Tim G. J. Rudner is funded by the Rhodes Trust and the EPSRC. We would like to thank Yarin Gal and Piotr Milo for helpful comments.

## Footnotes

[2]We use implementations provided by the authors `https://github.com/haarnoja/sac` for v1 and `https://github.com/vitchyr/rlkit` for v2.

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
