[Supplementary Material]

# A   A Brief Review of EM and Variational Inference

Fig. 5 shows the representation of a generative graphical model that produces observations $x$ from a distribution $x \sim p_\omega(x|h)$, has hidden variables $h$, and is parameterised by a set of parameters, $\omega$. In learning a model, we often seek the parameters that maximises the log-marginal-likelihood (LML), which can be found by marginalising the joint distribution $p_\omega(x, h)$ over hidden variables:

$$\ell_\omega(x) \coloneqq \log p_\omega(x) = \log\left(\int_{\mathcal{H}} p_\omega(x, h)dh\right). \qquad (8)$$

In many cases, we also need to infer the corresponding posterior,

$$p_\omega(h|x) = \frac{p_\omega(x, h)}{\int_{\mathcal{H}} p_\omega(x, h)dh}.$$

Figure 5: Graphical model of inference problem.

Evaluating the marginal likelihood in Eq. (8) and obtain the corresponding posterior, however, is intractable for most distributions. To compute the marginal likelihood and $\omega^*$, we can use the EM algorithm [13] and variational inference (VI). We review these two methods now.

For any valid probability distribution $q(h)$ with support over $h$ we can rewrite the LML as a difference of two divergences [30],

$$\ell_\omega(x) = \int_{\mathcal{H}} q(h) \log\left(\frac{p_\omega(x, h)}{q(h)}\right) dh - \int_{\mathcal{H}} q(h) \log\left(\frac{p_\omega(h|x)}{q(h)}\right) dh,$$
$$= \mathcal{L}(\omega, q(h)) + \mathrm{KL}(q(h) \parallel p_\omega(h|x)), \qquad (9)$$

where $\mathcal{L}(\omega, q(h)) \coloneqq \int_{\mathcal{H}} q(h) \log\left(\frac{p_\omega(x,h)}{q(h)}\right) dh$ is known as the evidence lower bound (ELBO). Intuitively, as $\mathrm{KL}(q(h) \parallel p_\omega(h|x)) \geq 0$, it follows that $\ell_\omega(x) \geq \mathrm{ELBO}\,(q(h); \omega)$, hence $\ell_\omega(x) \geq \mathrm{ELBO}\,(q(h); \omega)$ is a lower bound for the LML. The derivation of this bound can also be viewed as applying Jensen's inequality directly to Eq. (8) [8]. Note that when the ELBO and marginal likelihood are identical, the resulting KL divergence between the function $q(h)$ and the posterior $p(h|x)$ is zero, implying that $q(h) = p_\omega(h|x)$.

Maximising the LML now reduces to maximising the ELBO, which can be achieved iteratively using EM [13, 71]; an expectation step (E-step) finds the posterior for the current set of model parameters and then a maximisation step (M-step) maximises the ELBO with respect to $\omega$ while keeping $q(h)$ fixed as the posterior from the E-step.

As finding the exact posterior in the E-step is still typically intractable, we resort to variational inference (VI), a powerful tool for approximating the posterior using a parametrised variational distribution $q_\theta(h)$ [30, 4]. VI aims to reduce the KL divergence between the true posterior and the variational distribution, $\mathrm{KL}(q_\theta(h) \parallel p_\omega(h|x))$. Typically VI never brings this divergence to zero but nonetheless yields useful posterior approximations. As minimising $\mathrm{KL}(q_\theta(h) \parallel p_\omega(h|x))$ is equivalent to maximising the ELBO for the variational distribution (see Eq. (23) from Theorem 3), the variational E-step amounts to maximising the ELBO with respect to $\theta$ while keeping $\omega$ constant. The variational EM algorithm can be summarised as:

$$\text{Variational E-Step: } \theta_{k+1} \leftarrow \arg\max_\theta \mathcal{L}(\omega_k, \theta),$$
$$\text{Variational M-Step: } \omega_{k+1} \leftarrow \arg\max_\omega \mathcal{L}(\omega, \theta_{k+1}).$$

# B   A Probabilistic Interpretation of VIREL

We now motivate our inference procedure and Boltzmann distribution $\pi_\omega(a|s)$ from a probabilistic perspective, demonstrating that $\pi_\omega(a|s)$ can be interpreted as an action-posterior that characterises the uncertainty our model has in the optimality of $\hat{Q}_\omega(h)$. Moreover, maximising $\mathcal{L}(\omega, \theta)$ for $\theta$ is equivalent to carrying our variational inference on the graphical model in Fig. 6 for any $\varepsilon_\omega > 0$.

## B.1 Model Specification

Like previous work, we introduce a binary variable $\mathcal{O} \in \{0, 1\}$ in order to define a formal graphical model for our inference problem when $\varepsilon_\omega > 0$. The likelihood of $\mathcal{O}$ therefore takes the form of a Bernoulli distribution:

$$p_\omega(\mathcal{O}|h) = y_\omega(h)^{\mathcal{O}}(1 - y_\omega(h))^{(1-\mathcal{O})},$$

where

$$y_\omega(h) := \exp\left(\frac{\hat{Q}_\omega(h) - \max_{a'} \hat{Q}_\omega(a', s)}{\varepsilon_\omega}\right).$$

In most existing frameworks, $\mathcal{O} = 1$ is understood to be the event that the agent is acting optimally [35, 63]. As we are using function approximators in VIREL, $\mathcal{O} = 1$ can be interpreted as the event that the agent is behaving optimally under $\hat{Q}_\omega(h)$. Exploring the semantics of $\mathcal{O}$ further, consider the likelihood when $\mathcal{O} = 1$:

$$p_\omega(\mathcal{O} = 1|h) = \exp\left(\frac{\hat{Q}_\omega(h) - \max_{a'} \hat{Q}_\omega(a', s)}{\varepsilon_\omega}\right),$$

Observe that $0 \leq p_\omega(\mathcal{O} = 1|\cdot) \leq 1 \; \forall \; \omega \in \Omega \; s.t. \; \varepsilon_\omega > 0$. For any state $s$ and any action $a^*$ such that $p_\omega(\mathcal{O} = 1|s, a^*) = 1$, such an action must be optimal under $\hat{Q}_\omega(h)$ in the sense that it is the greedy action $a^* \in \arg\max_a \hat{Q}_\omega(h)$. If we find $p_\omega(\mathcal{O} = 1|h) = 1 \; \forall \; h \in \mathcal{H}$, then all observed state-action pairs have been generated from a greedy policy $\pi(a|s) = \delta(a \in \arg\max_{a'} \hat{Q}_\omega(a'|s))$. From Theorem 2, the closer the residual error $\varepsilon_\omega$ is to zero, the closer $\hat{Q}_\omega(h)$ becomes to representing an optimal action-value function. When $\varepsilon_\omega \approx 0$, any $a$ observed such that $p_\omega(\mathcal{O} = 1|a, \cdot) = 1$ will be very nearly an action sampled from an optimal policy, that is $a \sim \pi(a|\cdot) \approx \delta(a \in \arg\max_{a'} Q^*(a'|\cdot))$. We caution readers that in the limit $\varepsilon_\omega \to 0$, our likelihood is not well-defined for any

Figure 6: Graphical model for VIREL (variational approximation dashed)

$a \in \arg\max_{a'} \hat{Q}_\omega(a', s)$. Without loss of generality, we condition on optimality for the rest of this section, writing $\mathcal{O}$ in place of $\mathcal{O} = 1$. Defining the function $y_\omega(s) := \exp\left(-\frac{\max_{a'} \hat{Q}_\omega(a', s)}{\varepsilon_\omega}\right)$, our likelihood takes the convenient form:

$$p_\omega(\mathcal{O}|h) = \exp\left(\frac{\hat{Q}_\omega(h)}{\varepsilon_\omega}\right) y_\omega(s),$$

Defining the prior distribution as the uniform distribution $p(h) = \mathcal{U}(h)$ completes our model, the graph for which is shown in Fig. 6. Using Bayes' rule, we find our posterior distribution is:

$$
\begin{aligned}
p_\omega(h|\mathcal{O}) &= \frac{p_\omega(\mathcal{O}|h)p(h)}{p_\omega(\mathcal{O})}, \\
&= \frac{p_\omega(\mathcal{O}|h)p(h)}{\int_{\mathcal{H}} p_\omega(\mathcal{O}|h)p(h)dh}, \\
&= \frac{\exp\left(\frac{\hat{Q}_\omega(h)}{\varepsilon_\omega}\right) y_\omega(s)}{\int_{\mathcal{H}} \exp\left(\frac{\hat{Q}_\omega(h)}{\varepsilon_\omega}\right) y_\omega(s)dh}.
\end{aligned}
\tag{10}
$$

We can also derive our action-posterior, $p_\omega(a|s, \mathcal{O})$, which we will find to be equivalent to the Boltzmann policy from Eq. (3). Using Bayes' rule, it follows:

$$p_\omega(a|s, \mathcal{O}) = \frac{p_\omega(h|\mathcal{O})}{p_\omega(s|\mathcal{O})}.$$

Now, we find $p_\omega(s|\mathcal{O})$ by marginalising our posterior over actions. Substituting $p_\omega(s|\mathcal{O}) = \int p_\omega(h|\mathcal{O})da$ yields :

$$p_\omega(a|s, \mathcal{O}) = \frac{p_\omega(h|\mathcal{O})}{\int_{\mathcal{A}} p_\omega(h|\mathcal{O})da}.$$

Substituting for our posterior from Eq. (10), we obtain:

$$p_\omega(a|s, \mathcal{O}) = \frac{\exp\left(\frac{\hat{Q}_\omega(h)}{\varepsilon_\omega}\right) y_\omega(s)}{\int_\mathcal{A} \exp\left(\frac{\hat{Q}_\omega(h)}{\varepsilon_\omega}\right) y_\omega(s) da} \cdot \frac{\int_\mathcal{H} \exp\left(\frac{\hat{Q}_\omega(h)}{\varepsilon_\omega}\right) y_\omega(s) dh}{\int_\mathcal{H} \exp\left(\frac{\hat{Q}_\omega(h)}{\varepsilon_\omega}\right) y_\omega(s) dh},$$

$$= \frac{\exp\left(\frac{\hat{Q}_\omega(h)}{\varepsilon_\omega}\right) y_\omega(s)}{\left(\int_\mathcal{A} \exp\left(\frac{\hat{Q}_\omega(h)}{\varepsilon_\omega}\right) da\right) y_\omega(s)},$$

$$= \frac{\exp\left(\frac{\hat{Q}_\omega(h)}{\varepsilon_\omega}\right)}{\int_\mathcal{A} \exp\left(\frac{\hat{Q}_\omega(h)}{\varepsilon_\omega}\right) da},$$

$$= \pi_\omega(a|s),$$

proving that our action-posterior is exactly the Boltzmann policy introduced in Section 3.1. From a Bayesian perspective, the action-posterior $p_\omega(a|s, \mathcal{O})$ characterises the uncertainty we have in deducing the optimal action for a given state $s$ under $\hat{Q}_\omega(h)$; whenever $\varepsilon_\omega \approx 0$ and hence $\hat{Q}_\omega(h) \approx Q^*(h)$, the uncertainty will be very small as $p_\omega(a|s, \mathcal{O})$ will have near-zero variance, approximating a Dirac-delta distribution. Our model is therefore highly confident that the maximum-a-posteriori (MAP) action $a \in \arg\max_{a'} \hat{Q}_\omega(a', s)$ is an optimal action, with all of the probability mass being close to this point. In light of this, we can interpret the greedy policy $\pi_\omega(a|s) = \delta(a \in \arg\max_{a'} \hat{Q}_\omega(a', s))$ as one that always selecting the MAP action across all states.

As our model incorporates the uncertainty in the optimality of $\hat{Q}_\omega(h)$ into the variance of $\pi_\omega(a|s)$, we can benefit directly by sampling trajectories from $\pi_\omega(a|s)$ which drives exploration to gather data that is beneficial to reducing the residual error $\varepsilon_\omega$. Unfortunately, calculating the normalisation constant $\int_\mathcal{A} \exp\left(\frac{\hat{Q}_\omega(h)}{\varepsilon_\omega}\right) da$ is intractable for most function approximators and MDPs of interest. As such, we resort to variational inference, a powerful technique to infer an approximation to a posterior distribution from a tractable family of variational distributions [30, 4, 8]. As before $\pi_\theta(a|s)$ is known as the variational policy, is parametrised by $\theta \in \Theta$ and with the same support as $\pi_\omega(a|s)$. Like in Section 3.1, we define a variational distribution as $q_\theta(h) := d(s)\pi_\theta(a|s)$, where $d(s)$ is an arbitrary sampling distribution with support over $\mathcal{S}$. We fix $d(s)$, as in our model-free paradigm we do not learn the state transition dynamics and only seek to infer the action-posterior.

The goal of variational inference is to find $q_\theta(h)$ closest in KL-divergence to $p_\omega(h|\mathcal{O})$, giving an objective:

$$\theta^* \in \arg\min_\theta \mathrm{KL}(q_\theta(h) \parallel p_\omega(h|\mathcal{O})).$$

This objective still requires the intractable computation of $\int \exp\left(\frac{\hat{Q}_\omega(h)}{\varepsilon_\omega}\right) y_\omega(s) dh$. Using Eq. (9), we can overcome this by writing the KL divergence in terms of the ELBO:

$$\mathrm{KL}(q_\theta(h) \parallel p_\omega(h|\mathcal{O})) = \ell_\omega - \mathcal{L}_\omega(\theta),$$

where $\quad \ell_\omega := \log \int_\mathcal{H} \exp\left(\frac{\hat{Q}_\omega(h)}{\varepsilon_\omega}\right) y_\omega(s) dh, \quad \mathcal{L}_\omega(\theta) := \mathbb{E}_{h \sim q_\theta(h)}\left[\log\left(\frac{\exp\left(\frac{\hat{Q}_\omega(h)}{\varepsilon_\omega}\right) y_\omega(s)}{q_\theta(h)}\right)\right].$

We see that minimising the KL-divergence for $\theta$ is equivalent to maximising the ELBO for $\theta$, which is tractable. This affords a new objective:

$$\theta^* \in \arg\max_\theta \mathcal{L}_\omega(\theta).$$

Expanding the ELBO yields:

$$\mathcal{L}_\omega(\theta) = \mathbb{E}_{h \sim q_\theta(h)} \left[ \log \left( \frac{\exp\left(\frac{\hat{Q}_\omega(h)}{\varepsilon_\omega}\right) y_\omega(s)}{q_\theta(h)} \right) \right],$$

$$= \mathbb{E}_{s \sim d(s)} \left[ \mathbb{E}_{a \sim \pi_\theta(a|s)} \left[ \log \left( \frac{\exp\left(\frac{\hat{Q}_\omega(h)}{\varepsilon_\omega}\right) y_\omega(s)}{q_\theta(h)} \right) \right] \right],$$

$$= \mathbb{E}_{s \sim d(s)} \left[ \mathbb{E}_{a \sim \pi_\theta(a|s)} \left[ \frac{\hat{Q}_\omega(h)}{\varepsilon_\omega} \right] + \mathbb{E}_{a \sim \pi_\theta(a|s)} \left[ \log y_\omega(s) \right] - \mathbb{E}_{a \sim \pi_\theta(a|s)} \left[ \log(\pi_\theta(a|s)d(s)) \right] \right],$$

$$= \mathbb{E}_{s \sim d(s)} \left[ \mathbb{E}_{a \sim \pi_\theta(a|s)} \left[ \frac{\hat{Q}_\omega(h)}{\varepsilon_\omega} \right] + \log y_\omega(s) - \log d(s) - \mathbb{E}_{a \sim \pi_\theta(a|s)} \left[ \log \pi_\theta(a|s) \right] \right],$$

$$= \mathbb{E}_{s \sim d(s)} \left[ \mathbb{E}_{a \sim \pi_\theta(a|s)} \left[ \frac{\hat{Q}_\omega(h)}{\varepsilon_\omega} \right] - \mathbb{E}_{a \sim \pi_\theta(a|s)} \left[ \log \pi_\theta(a|s) \right] \right] + \mathbb{E}_{s \sim d(s)} \left[ \log \left( \frac{y_\omega(s)}{d(s)} \right) \right],$$

$$= \mathbb{E}_{s \sim d(s)} \left[ \mathbb{E}_{a \sim \pi_\theta(a|s)} \left[ \frac{\hat{Q}_\omega(h)}{\varepsilon_\omega} \right] + \mathscr{H}(\pi_\theta(a|s)) \right] + \mathbb{E}_{s \sim d(s)} \left[ \log \left( \frac{y_\omega(s)}{d(s)} \right) \right].$$

As the final term $\mathbb{E}_{s \sim d(s)} \left[ \log \left( \frac{y_\omega(s)}{d(s)} \right) \right]$ has no dependency on $\theta$, we can neglect it from our objective, recovering the VIREL objective from Eq. (4):

$$\mathcal{L}_\omega(\theta) = \mathbb{E}_{s \sim d(s)} \left[ \mathbb{E}_{a \sim \pi_\theta(a|s)} \left[ \frac{\hat{Q}_\omega(h)}{\varepsilon_\omega} \right] + \mathscr{H}(\pi_\theta(a|s)) \right].$$

Finally, Theorem 3 guarantees that minimising $\mathcal{L}_\omega(\theta)$ always minimises the expected KL divergence between $\pi_\omega(a|s)$ and $\pi_\theta(a|s)$, allowing us to learn a variational approximation for the action-posterior.

## C  A Discussion of the Target Set $\mathbb{T}$

We now prove that the Bellman operator for the Boltzmann policy, $\mathcal{T}^{\pi_\omega} \cdot := r(h) + \gamma \mathbb{E}_{h' \sim p(s'|h)\pi_\omega(a'|s')} [\cdot]$, is a member of $\mathbb{T}$. Taking the limit $\varepsilon_\omega \to 0$ of $\mathcal{T}^{\pi_\omega} \hat{Q}_\omega(h)$, we find:

$$\lim_{\varepsilon_\omega \to 0} \mathcal{T}^{\pi_\omega} \hat{Q}_\omega(h) = r(h) + \lim_{\varepsilon_\omega \to 0} \gamma \mathbb{E}_{h' \sim p(s'|h)\pi_\omega(a'|s')} \left[ \hat{Q}_\omega(h') \right].$$

From Theorem 2, evaluating $\lim_{\varepsilon_\omega \to 0} \gamma \mathbb{E}_{h' \sim p(s'|h)\pi_\omega(a'|s')} [\cdot]$ recovers a Dirac-delta distribution:

$$\lim_{\varepsilon_\omega \to 0} \mathcal{T}^{\pi_\omega} \hat{Q}_\omega(h) = r(h) + \gamma \mathbb{E}_{h' \sim p(s'|h)\delta(a' = \arg\max_a \hat{Q}_\omega(a,s))} \left[ \hat{Q}_\omega(h') \right],$$

$$= r(h) + \gamma \mathbb{E}_{h' \sim p(s'|h)} \left[ \max_{a'} (\hat{Q}_\omega(h')) \right],$$

$$= \mathcal{T}^* \hat{Q}_\omega(h).$$

which is sufficient to demonstrate membership of $\mathbb{T}$.

Observe that using $\mathcal{T}^{\pi_\omega} \cdot$ implies $\hat{Q}_\omega(h)$ cannot represent the true $Q$-function of any $\pi_\omega(a|s)$ except for the optimal $Q$-function. To see this, imagine there exists some $\varepsilon_\omega > 0$ such that $Q^{\pi_\omega}(\cdot) = \hat{Q}(\cdot)$. Under these conditions, it holds that $\mathcal{T}^{\pi_\omega} \hat{Q}(\cdot) = \hat{Q}(\cdot) \implies \varepsilon_\omega = 0$, which is a contradiction. More generally, as $\pi_\omega(a|s)$ is defined in terms of $\varepsilon_\omega$, which itself depends on $\pi_\omega(a|s)$ from the definition of $\mathcal{T}^{\pi_\omega} \cdot$, any $\omega$ satisfying this recursive definition forms a constrained set $\Omega^c \subseteq \Omega$. Crucially, we show in Theorem 2 that there always exists some $\omega^* \in \Omega^c$ such that $\hat{Q}_{\omega^*}$ can represent the action-value function for an optimal policy. Note that there may exist other policies that are not Boltzmann distributions such that $\hat{Q}_\omega(h) = Q^\pi(h)$ for some $\omega \in \Omega^c$. We discuss operators that don't constrain $\Omega$ in Appendix F.2.

Finally, we can approximate $\mathcal{T}^{\pi_\omega}$ using any TD target sampled from $\pi_\omega(a|s)$ (see Sutton & Barto [55] for an overview of TD methods). Likewise, the optimum Bellman operator $\mathcal{T}^* \cdot = r(h) + \gamma \mathbb{E}_{h' \sim p(s'|h)} [\max_{a'}(\cdot)]$ is by definition a member of $\mathbb{T}$ and can be approximated using the Q-learning target [68].

## D  Proofs for Section 3

### D.1  Derivation of Lower Bound in terms of KL Divergence

We need to show that

$$\mathcal{L}(\omega, \theta) = \ell(\omega) - \mathrm{KL}(q_\theta(h) \parallel p_\omega(h)) - \mathscr{H}(d(s)), \qquad (11)$$

$$\text{where} \quad \ell(\omega) := \log \int_{\mathcal{H}} \exp\left(\frac{\hat{Q}_\omega(h)}{\varepsilon_\omega}\right) dh, \quad p_\omega(h) := \frac{\exp\left(\frac{\hat{Q}_\omega(h)}{\varepsilon_\omega}\right)}{\int_{\mathcal{H}} \exp\left(\frac{\hat{Q}_\omega(h)}{\varepsilon_\omega}\right) dh}.$$

Starting with the LHS of Eq. (11), and recalling the definition of $\mathcal{L}(\omega, \theta)$ from Eq. (4), we have:

$$\mathcal{L}(\omega, \theta) = \mathbb{E}_{s \sim d(s)}\left[ \mathbb{E}_{a \sim \pi_\theta(a|s)}\left[ \frac{\hat{Q}_\omega(h)}{\varepsilon_\omega} \right] + \mathscr{H}(\pi_\theta(a|s)) \right].$$

Expanding the definition of differential entropy, $\mathscr{H}(\pi_\theta(a|s))$:

$$\mathcal{L}(\omega, \theta) = \mathbb{E}_{s \sim d(s)}\left[ \mathbb{E}_{a \sim \pi_\theta(a|s)}\left[ \frac{\hat{Q}_\omega(h)}{\varepsilon_\omega} \right] - \mathbb{E}_{a \sim \pi_\theta(a|s)}\left[ \log \pi_\theta(a|s) \right] \right],$$

$$= \mathbb{E}_{s \sim d(s)}\left[ \mathbb{E}_{a \sim \pi_\theta(a|s)}\left[ \frac{\hat{Q}_\omega(h)}{\varepsilon_\omega} \right] - \mathbb{E}_{a \sim \pi_\theta(a|s)}\left[ \log\left( \frac{\pi_\theta(a|s)d(s)}{p_\omega(h)} \cdot \frac{p_\omega(h)}{d(s)} \right) \right] \right],$$

$$= \mathbb{E}_{s \sim d(s)}\left[ \mathbb{E}_{a \sim \pi_\theta(a|s)}\left[ \frac{\hat{Q}_\omega(h)}{\varepsilon_\omega} \right] - \mathbb{E}_{a \sim \pi_\theta(a|s)}\left[ \log\left( \frac{q_\theta(h)}{p_\omega(h)} \right) \right] \right.$$

$$\left. - \mathbb{E}_{a \sim \pi_\theta(a|s)}\left[ \log p_\omega(h) \right] + \log(d(s)) \right],$$

$$= \mathbb{E}_{h \sim q_\theta(h)}\left[ \frac{\hat{Q}_\omega(h)}{\varepsilon_\omega} \right] - \mathrm{KL}(q_\theta(h) \parallel p_\omega(h)) - \mathscr{H}(d(s)) - \mathbb{E}_{h \sim q_\theta(h)}\left[ \log p_\omega(h) \right].$$

Substituting for the definition of $p_\omega(h)$ in the final term yields our desired result:

$$\mathcal{L}(\omega, \theta) = \mathbb{E}_{h \sim q_\theta(h)}\left[ \frac{\hat{Q}_\omega(h)}{\varepsilon_\omega} \right] - \mathrm{KL}(\pi_\theta(a|s) \parallel \pi_\omega(a|s)) - \mathscr{H}(d(s))$$

$$- \mathbb{E}_{a \sim \pi_\theta(a|s)}\left[ \log\left( \frac{\exp\left(\frac{\hat{Q}_\omega(h)}{\varepsilon_\omega}\right)}{\int_{\mathcal{H}} \exp\left(\frac{\hat{Q}_\omega(h)}{\varepsilon_\omega}\right) dh} \right) \right],$$

$$= \mathbb{E}_{h \sim q_\theta(h)}\left[ \frac{\hat{Q}_\omega(h)}{\varepsilon_\omega} \right] - \mathrm{KL}(\pi_\theta(a|s) \parallel \pi_\omega(a|s)) - \mathscr{H}(d(s))$$

$$- \mathbb{E}_{h \sim q_\theta(h)}\left[ \frac{\hat{Q}_\omega(h)}{\varepsilon_\omega} \right] + \log \int_{\mathcal{H}} \exp\left(\frac{\hat{Q}_\omega(h)}{\varepsilon_\omega}\right) dh,$$

$$= \ell(\omega) - \mathrm{KL}(q_\theta(h) \parallel p_\omega(h)) - \mathscr{H}(d(s)).$$

## D.2 Convergence of Boltzmann Distribution to Dirac-Delta

**Theorem 1** (Convergence of Boltzmann Distribution to Dirac Delta). *Let $p_\varepsilon : \mathcal{X} \to [0,1]$ be a Boltzmann distribution with temperature $\varepsilon \in \mathbb{R}_{\geq 0}$*

$$p_\varepsilon(x) = \frac{\exp\left(\frac{f(x)}{\varepsilon}\right)}{\int_{\mathcal{X}} \exp\left(\frac{f(x)}{\varepsilon}\right) dx},$$

*where $f : \mathcal{X} \to \mathcal{Y}$ is a function with a unique maximum $f(x^*) = \sup_x f$, a compact domain $\mathcal{X}$ and bounded range $\mathcal{Y}$. Let $f$ be locally $\mathbb{C}^2$ smooth about $x^*$, that is $\exists \Delta > 0$ s.t. $f(x) \in \mathbb{C}^2 \ \forall \ x \in \{x | \|x - x^*\| < \Delta \}$. In the limit $\varepsilon \to 0$, $p_\varepsilon(x) \to \delta(x^*)$, that is:*

$$\lim_{\varepsilon \to 0} \int_{\mathcal{X}} \varphi(x) p_\varepsilon(x) dx = \varphi(x^*), \tag{12}$$

*for any smooth test function $\varphi \in \mathbb{C}_0^\infty(\mathcal{X})$.*

*Proof.* Firstly, we define the auxiliary function to be

$$g(x) := f(x) - f(x^*).$$

Note, $g(x) \leq 0$ with equality at $g(x^*) = 0$. Substituting $f(x) = g(x) + f(x^*)$ into $p_\varepsilon(x)$:

$$
\begin{aligned}
p_\varepsilon(x) &= \frac{\exp\left(\frac{g(x)+f(x^*)}{\varepsilon}\right)}{\int_{\mathcal{X}} \exp\left(\frac{g(x)+f(x^*)}{\varepsilon}\right) dx}, \\
&= \frac{\exp\left(\frac{g(x)}{\varepsilon}\right) \exp\left(\frac{f(x^*)}{\varepsilon}\right)}{\int_{\mathcal{X}} \exp\left(\frac{g(x)}{\varepsilon}\right) \exp\left(\frac{f(x^*)}{\varepsilon}\right) dx}, \\
&= \frac{\exp\left(\frac{g(x)}{\varepsilon}\right)}{\int_{\mathcal{X}} \exp\left(\frac{g(x)}{\varepsilon}\right) dx}.
\end{aligned}
\tag{13}
$$

Now, substituting Eq. (13) into the limit in Eq. (12) yields:

$$\lim_{\varepsilon \to 0} \int_{\mathcal{X}} \varphi(x) p_\varepsilon(x) dx = \lim_{\varepsilon \to 0} \left( \int_{\mathcal{X}} \varphi(x) \frac{\exp\left(\frac{g(x)}{\varepsilon}\right)}{\int_{\mathcal{X}} \exp\left(\frac{g(x)}{\varepsilon}\right) dx} dx \right). \tag{14}$$

Using the substitution $u := \frac{(x^* - x)}{\sqrt{\varepsilon}}$ to transform the integrals in Eq. (14), we obtain

$$
\begin{aligned}
\lim_{\varepsilon \to 0} \int_{\mathcal{X}} \varphi(x) p_\varepsilon(x) dx &= \lim_{\varepsilon \to 0} \left( \int_{\mathcal{U}} \varphi(x^* - \sqrt{\varepsilon} u) \frac{\exp\left(\frac{g(x^* - \sqrt{\varepsilon} u)}{\varepsilon}\right)}{\int_{\mathcal{U}} \exp\left(\frac{g(x^* - \sqrt{\varepsilon} u)}{\varepsilon}\right) \sqrt{\varepsilon} du} \sqrt{\varepsilon} du \right), \\
&= \lim_{\varepsilon \to 0} \left( \frac{\int_{\mathcal{U}} \varphi(x^* - \sqrt{\varepsilon} u) \exp\left(\frac{g(x^* - \sqrt{\varepsilon} u)}{\varepsilon}\right) du}{\int_{\mathcal{U}} \exp\left(\frac{g(x^* - \sqrt{\varepsilon} u)}{\varepsilon}\right) du} \right).
\end{aligned}
\tag{15}
$$

We now find $\lim_{\varepsilon \to 0} \left( \frac{g(x^* - \sqrt{\varepsilon}u)}{\varepsilon} \right)$. Denoting the partial derivative $\partial_{\sqrt{\varepsilon}} := \frac{\partial}{\partial \sqrt{\varepsilon}}$ and using L'Hôpital's rule to the second derivative with respect to $\sqrt{\epsilon}$, we find the limit as:

$$
\begin{aligned}
\lim_{\varepsilon \to 0} \left( \frac{g(x^* - \sqrt{\varepsilon}u)}{\varepsilon} \right) &= \lim_{\varepsilon \to 0} \left( \frac{\partial_{\sqrt{\varepsilon}} g(x^* - \sqrt{\varepsilon}u)}{\partial_{\sqrt{\varepsilon}} \varepsilon} \right), \\
&= \lim_{\varepsilon \to 0} \left( \frac{\partial_{\sqrt{\varepsilon}} f(x^* - \sqrt{\varepsilon}u)}{\partial_{\sqrt{\varepsilon}} \varepsilon} \right), \\
&= \lim_{\varepsilon \to 0} \left( \frac{-u^\top \nabla f(x^* - \sqrt{\varepsilon}u)}{2\sqrt{\varepsilon}} \right), \\
&= \lim_{\varepsilon \to 0} \left( \frac{-\partial_{\sqrt{\varepsilon}} \left( u^\top \nabla f(x^* - \sqrt{\varepsilon}u) \right)}{\partial_{\sqrt{\varepsilon}} (2\sqrt{\varepsilon})} \right), \\
&= \lim_{\varepsilon \to 0} \left( \frac{u^\top \nabla^2 f(x^* - \sqrt{\varepsilon}u)u}{2} \right), \\
&= \frac{u^\top \nabla^2 f(x^*)u}{2}.
\end{aligned}
$$

The integrand in the numerator in Eq. (15) therefore converges pointwise to $\varphi(x^*) \exp\left( \frac{u^\top \nabla^2 f(x^*)u}{2} \right)$, that is

$$
\lim_{\varepsilon \to 0} \left( \varphi(x^* - \sqrt{\varepsilon}u) \exp\left( \frac{g(x^* - \sqrt{\varepsilon}u)}{\varepsilon} \right) \right) = \varphi(x^*) \exp\left( \frac{u^\top \nabla^2 f(x^*)u}{2} \right), \tag{16}
$$

and the integrand in the denominator converges pointwise to $\exp\left( \frac{u^\top \nabla^2 f(x^*)u}{2} \right)$, that is

$$
\lim_{\varepsilon \to 0} \left( \exp\left( \frac{g(x^* - \sqrt{\varepsilon}u)}{\varepsilon} \right) \right) = \exp\left( \frac{u^\top \nabla^2 f(x^*)u}{2} \right). \tag{17}
$$

From the second order sufficient conditions for $f(x^*)$ to be a maximum, we have $u^\top \nabla^2 f(x^*)u \le 0$ $\forall \, u \in \mathcal{U}$ with equality only when $u = 0$ [37]. This implies that Eq. (16) and Eq. (17) are both bounded functions.

By definition, we have $g(x^* - \sqrt{\epsilon}u) \le 0 \, \forall \, u \in \mathcal{U}$, which implies that $\left| \exp\left( \frac{g(x^* - \sqrt{\varepsilon}u)}{\varepsilon} \right) \right| \le 1$. Consequently, the integrand in the numerator of Eq. (15) is dominated by $\|\varphi(\cdot)\|_\infty$, that is

$$
\left| \varphi(x^* - \sqrt{\varepsilon}u) \exp\left( \frac{g(x^* - \sqrt{\varepsilon}u)}{\varepsilon} \right) \right| \le \|\varphi(\cdot)\|_\infty, \tag{18}
$$

and the integrand in the denominator is dominated by 1, that is

$$
\left| \exp\left( \frac{g(x^* - \sqrt{\varepsilon}u)}{\varepsilon} \right) \right| \le 1. \tag{19}
$$

Together Eqs. (16) to (19) are the sufficient conditions for applying the dominated convergence theorem [3], allowing us to commute all limits and integrals in Eq. (15), yielding our desired result:

$$
\begin{aligned}
\lim_{\varepsilon \to 0} \int_{\mathcal{X}} \varphi(x) p_\varepsilon(x) dx &= \lim_{\varepsilon \to 0} \left( \frac{\int_{\mathcal{U}} \varphi(x^* - \sqrt{\varepsilon}u) \exp\left( \frac{g(x^* - \sqrt{\varepsilon}u)}{\varepsilon} \right) du}{\int_{\mathcal{U}} \exp\left( \frac{g(x^* - \sqrt{\varepsilon}u)}{\varepsilon} \right) du} \right), \\
&= \frac{\int_{\mathcal{U}} \lim_{\varepsilon \to 0} \left( \varphi(x^* - \sqrt{\varepsilon}u) \exp\left( \frac{g(x^* - \sqrt{\varepsilon}u)}{\varepsilon} \right) \right) du}{\int_{\mathcal{U}} \lim_{\varepsilon \to 0} \left( \exp\left( \frac{g(x^* - \sqrt{\varepsilon}u)}{\varepsilon} \right) \right) du}, \\
&= \frac{\int_{\mathcal{U}} \varphi(x^*) \exp\left( u^\top \nabla^2 f(x^*)u \right) du}{\int_{\mathcal{U}} \exp\left( u^\top \nabla^2 f(x^*)u \right) du}, \\
&= \varphi(x^*) \frac{\int_{\mathcal{U}} \exp\left( u^\top \nabla^2 f(x^*)u \right) du}{\int_{\mathcal{U}} \exp\left( u^\top \nabla^2 f(x^*)u \right) du}, \\
&= \varphi(x^*).
\end{aligned}
$$

$\square$

## D.3 Optimal Boltzmann Distributions as Optimal Policies

**Lemma 1** (Lower and Upper limits of $\mathcal{L}(\omega, \theta)$). *i) For any $\varepsilon_\omega > 0$ and $\pi_\theta(a|s) = \delta(a^*)$, we have $\mathcal{L}(\omega, \theta) = -\infty$. ii) For $\hat{Q}_\omega(\cdot) > 0$ and any non-deterministic $\pi_\theta(a|s)$, $\lim_{\varepsilon_\omega \to 0} \mathcal{L}(\omega, \theta) = \infty$.*

*Proof.* To prove i), we substitute $\pi_\theta(a|s) = \delta(a^*)$ into $\mathcal{L}(\omega, \theta)$ from Eq. (4), yielding:

$$\mathcal{L}(\omega, \theta) = \mathbb{E}_{s \sim d(s)} \left[ \mathbb{E}_{a \sim \delta(a^*)} \left[ \frac{\hat{Q}_\omega(h)}{\varepsilon_\omega} \right] + \mathcal{H}(\delta(a^*)) \right],$$

$$= \mathbb{E}_{s \sim d(s)} \left[ \frac{\hat{Q}_\omega(a^*, s)}{\varepsilon_\omega} + \mathcal{H}(\delta(a^*)) \right], \tag{20}$$

We now prove that $\mathcal{H}(\delta(a^*)) = -\infty$ for any $a^*$. Let $p : \mathcal{X} \to [0, 1]$ be any zero-mean, unit variance distribution. Using a transformation of variables, we have $\mathcal{A} = \sigma \mathcal{X} + a^*$ and hence $p(a) = \frac{1}{\sigma} p(\sigma x - a^*)$. We can therefore write our Dirac-delta distribution as

$$\delta(a^*) = \lim_{\sigma \to 0} p(a) = \lim_{\sigma \to 0} \frac{1}{\sigma} p(\sigma x - a^*).$$

Substituting into the definition of differential entropy, we obtain:

$$
\begin{aligned}
\mathcal{H}(\delta(a^*)) &= \lim_{\sigma \to 0} \mathcal{H}(p(a)) \\
&= \lim_{\sigma \to 0} \mathcal{H}\left( \frac{1}{\sigma} p(\sigma x - a^*) \right), \\
&= - \lim_{\sigma \to 0} \int_{\mathcal{A}} \frac{1}{\sigma} p(\sigma x - a^*) \log \left( \frac{1}{\sigma} p(\sigma x - a^*) \right) da, \\
&= - \lim_{\sigma \to 0} \int_{\mathcal{A}} \frac{1}{\sigma} p(\sigma x - a^*) \log \left( p(\sigma x - a^*) \right) da + \lim_{\sigma \to 0} \int_{\mathcal{A}} \frac{1}{\sigma} p(\sigma x - a^*) \log (\sigma) da, \\
&= - \int_{\mathcal{A}} \delta(a^*) \log (p(-a^*)) da + \lim_{\sigma \to 0} \log (\sigma), \\
&= - \log(p(-a^*)) + \lim_{\sigma \to 0} \log (\sigma), \\
&= -\infty.
\end{aligned}
\tag{21}
$$

Substituting for $\mathcal{H}(\delta(a^*))$ from Eq. (21) in Eq. (20) yields our desired result:

$$
\begin{aligned}
\mathcal{L}(\omega, \theta) &= \mathbb{E}_{s \sim d(s)} \left[ \frac{\hat{Q}_\omega(a^*, s)}{\varepsilon_\omega} \right] + \mathbb{E}_{s \sim d(s)} \left[ \mathcal{H}(\delta(a^*)) \right], \\
&= \frac{\mathbb{E}_{s \sim d(s)} \left[ \hat{Q}_\omega(a^*, s) \right]}{\varepsilon_\omega} + (\lim_{\sigma \to 0} \log (\sigma) - \log(p(-a^*))) \mathbb{E}_{s \sim d(s)} [1], \\
&= -\infty,
\end{aligned}
$$

where our final line follows from the first term being finite for any $\varepsilon_\omega > 0$.

To prove ii), we take the limit $\varepsilon_\omega \to 0$ of $\mathcal{L}(\omega, \theta)$ in Eq. (4):

$$
\begin{aligned}
\lim_{\varepsilon_\omega \to 0} \mathcal{L}(\omega, \theta) &= \lim_{\varepsilon_\omega \to 0} \left( \frac{\mathbb{E}_{d(s)\pi_\theta(a|s)} \left[ \hat{Q}_\omega(h) \right]}{\varepsilon_\omega} + \mathbb{E}_{d(s)} \left[ \mathcal{H}(\pi_\theta(a|s)) \right] \right), \\
&= \lim_{\varepsilon_\omega \to 0} \left( \frac{\mathbb{E}_{d(s)\pi_\theta(a|s)} \left[ \hat{Q}_\omega(h) \right]}{\varepsilon_\omega} \right) + \mathbb{E}_{d(s)} \left[ \mathcal{H}(\pi_\theta(a|s)) \right], \\
&= \infty.
\end{aligned}
$$

where our last line follows from $\mathscr{H}(\pi_\theta(a|s))$ being finite for any non-deterministic $\pi_\theta(a|s)$ and $\hat{Q}_\omega(\cdot) > 0 \implies \mathbb{E}_{d(s)\pi_\theta(a|s)}\left[\hat{Q}_\omega(h)\right] > 0.$

$\square$

**Theorem 2** (Optimal Boltzmann Distributions as Optimal Policies). *For any pair* $\{\omega^*, \theta^*\}$ *that maximises* $\mathcal{L}(\omega, \theta)$ *defined in Eq.* (4), *the corresponding variational policy induced must be optimal, i.e.* $\{\omega^*, \theta^*\} \in \arg\max_{\omega,\theta} \mathcal{L}(\omega, \theta) \implies \pi_{\omega^*}(a|s) \in \Pi^*$. *Moreover, any* $\theta^*$ *s.t.* $\pi_{\theta^*}(a|s) = \pi_{\omega^*}(a|s) \implies \theta^* \in \arg\max_{\omega,\theta} \mathcal{L}(\omega, \theta)$.

*Proof.* Our proof is structured as follows: Firstly, we prove that $\varepsilon_{\omega^*} = 0$ is both a necessary and sufficient condition for any $\omega^* \in \arg\max_{\omega,\theta} \mathcal{L}(\omega, \theta)$ with $\hat{Q}_{\omega^*}(\cdot) > 0$. We then verify that $\hat{Q}_{\omega^*}(\cdot) > 0$ is satisfied by our framework and $\varepsilon_{\omega^*} = 0$ is feasible. Finally, we prove that $\varepsilon_{\omega^*} = 0$ is sufficient for $\pi_{\omega^*}(a|s) \in \Pi^*$.

To prove necessity, assume there exists an optimal $\omega^*$ such that $\varepsilon_{\omega^*} \neq 0$. As $\varepsilon_\omega \geq 0$, it must be that $\varepsilon_{\omega^*} > 0$. Consider $\mathcal{L}(\omega, \theta)$ as defined in Eq. (4):

$$\mathcal{L}(\omega, \theta) = \frac{\mathbb{E}_{d(s)\pi_\theta(a|s)}\left[\hat{Q}_\omega(h)\right]}{\varepsilon_\omega} + \mathbb{E}_{d(s)}\left[\mathscr{H}(\pi_\theta(a|s))\right].$$

As $\pi_\theta(a|s)$ has finite variance, $\mathscr{H}(\pi_\theta(a|s))$ is upper bounded, and as $\hat{Q}_\omega(\cdot)$ is upper bounded, $\mathbb{E}_{d(s)\pi_\theta(a|s)}\left[\hat{Q}_\omega(h)\right]$ is upper bounded too. Together, this implies that $\mathbb{E}_{d(s)\pi_\theta(a|s)}\left[\hat{Q}_\omega(h)\right]$ is upper bounded for $\varepsilon_{\omega^*} > 0$. From Assumption 2, there exists $\omega^\diamond \in \Omega$ such that $\varepsilon_{\omega^\diamond} = 0$. From Lemma 1, there exists $\theta^*$ such that $\lim_{\varepsilon_{\omega^*} \to 0} \mathcal{L}(\omega^\diamond, \theta^*) = \infty$, implying $\mathcal{L}(\omega^*, \theta^*) < \mathcal{L}(\omega^\diamond, \theta^*)$ which is a contradiction.

To prove sufficiency, we take $\arg\max_\omega \mathcal{L}(\omega, \theta)$:

$$\arg\max_\omega \mathcal{L}(\omega, \theta) = \arg\max_\omega \left( \mathbb{E}_{d(s)\pi_\theta(a|s)}\left[\frac{\hat{Q}_\omega(h)}{\varepsilon_\omega}\right] + \mathbb{E}_{d(s)}\left[\mathscr{H}(\pi_\theta(a|s))\right] \right),$$

$$= \arg\max_\omega \left( \mathbb{E}_{d(s)\pi_\theta(a|s)}\left[\frac{\hat{Q}_\omega(h)}{\varepsilon_\omega}\right] \right),$$

$$= \arg\max_\omega \left( \frac{\mathbb{E}_{d(s)\pi_\theta(a|s)}\left[\hat{Q}_\omega(h)\right]}{\varepsilon_\omega} \right).$$

Assume that ① $\hat{Q}_{\omega^*}(\cdot) > 0$. It then follows:

$$\arg\max_\omega \mathcal{L}(\omega, \theta) = \arg\max_\omega \left( \frac{\mathbb{E}_{d(s)\pi_\theta(a|s)}\left[\hat{Q}_\omega(h)\right]}{\varepsilon_\omega} \right),$$

$$= \arg\min_\omega \left( \frac{\varepsilon_\omega}{\mathbb{E}_{d(s)\pi_\theta(a|s)}\left[\hat{Q}_\omega(h)\right]} \right),$$

$$= \arg\min_\omega \varepsilon_\omega,$$

which, as $\varepsilon_\omega \geq 0$, is satisfied for any $\omega^* \in \Omega$ s.t. $\varepsilon_{\omega^*} = 0$, proving sufficiency.

Assume now ② $\hat{Q}_{\omega^*}(\cdot)$ is locally smooth with a unique maximum over actions according to Definition 1. Under this condition we can apply Theorem 1 and our Boltzmann distribution tends towards a

Dirac-delta function:

$$\pi_{\omega^*}(a|s) = \lim_{\varepsilon_\omega \to 0} \left( \frac{\exp\left(\frac{\hat{Q}_{\omega^*}(h)}{\varepsilon_\omega}\right)}{\int \exp\left(\frac{\hat{Q}_{\omega^*}(h)}{\varepsilon_\omega}\right) da} \right) = \delta(a = \arg\max_{a'} \hat{Q}_{\omega^*}(s, a')), \tag{22}$$

which is a greedy policy w.r.t. $\hat{Q}_{\omega^*}(\cdot)$. From Definition 2, when $\lim_{\varepsilon_\omega \to 0} \pi_\omega(a|s)$ we have $\mathcal{T}_\omega \hat{Q}_\omega(h) = \mathcal{T}^* \hat{Q}_\omega(h)$. Substituting into $\varepsilon_{\omega^*} = 0$ shows our our function approximator must satisfy an optimal Bellman equation:

$$\varepsilon_{\omega^*} = \frac{c}{p}\|\mathcal{T}^* \hat{Q}_\omega(h) - \hat{Q}_\omega(h)\|_p^p = 0,$$
$$\implies \mathcal{T}^* \hat{Q}_{\omega^*}(\cdot) = \hat{Q}_{\omega^*}(\cdot),$$

hence $\hat{Q}_{\omega^*}(\cdot) = Q^*(\cdot)$. Under Assumption 2, we see that there exists $\omega^* \in \Omega$ $s.t.$ $\varepsilon_{\omega^*} = 0$ for $\hat{Q}_{\omega^*}(\cdot) = Q^*(\cdot)$, hence $\varepsilon_{\omega^*} = 0$ is feasible. Moreover, our assumptions ⓘ and ⓘⓘ are satisfied for $\hat{Q}_{\omega^*}(\cdot) = Q^*(\cdot)$ under Assumptions 2 and 3 respectively. Substituting for $\hat{Q}_{\omega^*}(\cdot) = Q^*(\cdot)$ into $\pi_{\omega^*}(a|s)$ from Eq. (22) we recover our desired result:

$$\omega^* \in \arg\max_\omega \mathcal{L}(\omega, \theta)$$
$$\implies \pi_{\omega^*}(a|s) = \delta(a = \arg\max_{a'} Q^*(s, a')) \in \Pi^*.$$

From Lemma 1, we have that $\mathcal{L}(\omega, \theta) \to \infty = \max_{\omega, \theta} \mathcal{L}(\omega, \theta)$ when $\varepsilon_\omega = 0$ for any $\theta^* \in \Theta$ such that the variational policy is non-deterministic, hence

$$\{\omega^*, \theta^*\} \in \arg\max_{\omega, \theta} \mathcal{L}(\omega, \theta) \implies \pi_{\omega^*}(a|s) \in \Pi^*,$$

as required. □

### D.4 Maximising the ELBO for $\theta$

**Theorem 3** (Maximising the ELBO for $\theta$). *Maximising $\mathcal{L}(\omega, \theta)$ for $\theta$ with $\varepsilon_\omega > 0$ is equivalent to minimising the expected KL divergence between $\pi_\omega(a|s)$ and $\pi_\theta(a|s)$, i.e. for any $\varepsilon_\omega > 0$, $\max_\theta \mathcal{L}(\omega, \theta) = \min_\theta \mathbb{E}_{d(s)} [\text{KL}(\pi_\theta(a|s) \| \pi_\omega(a|s))]$ with $\pi_\omega(a|s) = \pi_\theta(a|s)$ under exact representability.*

*Proof.* Firstly, we write $\mathcal{L}(\omega, \theta)$ in terms of $\ell(\omega)$ and $\text{KL}(q_\theta(h) \| p_\omega(h))$ from Eq. (5), ignoring the entropy term which has no dependency on $\omega$ and $\theta$:

$$\mathcal{L}(\omega, \theta) = \ell(\omega) - \text{KL}(q_\theta(h) \| p_\omega(h)),$$

which, for any $\varepsilon_\omega > 0$, implies

$$\max_\theta \mathcal{L}(\omega, \theta) = \max_\theta \left( \ell(\omega) - \text{KL}(q_\theta(h) \| p_\omega(h)) \right).$$
$$= \min_\theta \left( \text{KL}(q_\theta(h) \| p_\omega(h)) \right). \tag{23}$$

We now introduce the definition

$$p_\omega(s) := \frac{\int_{\mathcal{A}} \exp\left(\frac{\hat{Q}_\omega(h)}{\varepsilon_\omega}\right) da}{\int_{\mathcal{H}} \exp\left(\frac{\hat{Q}_\omega(h)}{\varepsilon_\omega}\right) dh}.$$

We now show that we can decompose $p_\omega(h)$ as $p_\omega(h) = \pi_\omega(a|s)p_\omega(s)$:

$$p_\omega(h) = \frac{\exp\left(\frac{\hat{Q}_\omega(h)}{\varepsilon_\omega}\right)}{\int_{\mathcal{H}} \exp\left(\frac{\hat{Q}_\omega(h)}{\varepsilon_\omega}\right) dh},$$

$$= \frac{\exp\left(\frac{\hat{Q}_\omega(h)}{\varepsilon_\omega}\right)}{\int_{\mathcal{H}} \exp\left(\frac{\hat{Q}_\omega(h)}{\varepsilon_\omega}\right) dh} \cdot \frac{\int_{\mathcal{A}} \exp\left(\frac{\hat{Q}_\omega(h)}{\varepsilon_\omega}\right) da}{\int_{\mathcal{A}} \exp\left(\frac{\hat{Q}_\omega(h)}{\varepsilon_\omega}\right) da},$$

$$= \frac{\exp\left(\frac{\hat{Q}_\omega(h)}{\varepsilon_\omega}\right)}{\int_{\mathcal{A}} \exp\left(\frac{\hat{Q}_\omega(h)}{\varepsilon_\omega}\right) da} \cdot \frac{\int_{\mathcal{A}} \exp\left(\frac{\hat{Q}_\omega(h)}{\varepsilon_\omega}\right) da}{\int_{\mathcal{H}} \exp\left(\frac{\hat{Q}_\omega(h)}{\varepsilon_\omega}\right) dh},$$

$$= \pi_\omega(a|s)p_\omega(s).$$

Substituting for $p_\omega(h) = \pi_\omega(a|s)p_\omega(s)$ and $q_\theta(h) = d(s)\pi_\theta(a|s)$ into the KL divergence from Eq. (23) yields:

$$\mathrm{KL}(q_\theta(h) \parallel p_\omega(h)) = \mathbb{E}_{d(s)\pi_\theta(a|s)}\left[\log\left(\frac{d(s)\pi_\theta(a|s)}{p_\omega(s)\pi_\omega(a|s)}\right)\right],$$

$$= \mathbb{E}_{d(s)\pi_\theta(a|s)}\left[\log\left(\frac{d(s)}{p_\omega(s)}\right)\right] + \mathbb{E}_{d(s)\pi_\theta(a|s)}\left[\log\left(\frac{\pi_\theta(a|s)}{\pi_\omega(a|s)}\right)\right],$$

$$= \mathbb{E}_{d(s)}\left[\log\left(\frac{d(s)}{p_\omega(s)}\right)\right]\mathbb{E}_{\pi_\theta(a|s)}[1] + \mathbb{E}_{d(s)\pi_\theta(a|s)}\left[\log\left(\frac{\pi_\theta(a|s)}{\pi_\omega(a|s)}\right)\right],$$

$$= \mathbb{E}_{d(s)}\left[\log\left(\frac{d(s)}{p_\omega(s)}\right)\right] + \mathbb{E}_{d(s)}\left[\mathbb{E}_{\pi_\theta(a|s)}\left[\log\left(\frac{\pi_\theta(a|s)}{\pi_\omega(a|s)}\right)\right]\right],$$

$$= \mathrm{KL}(d(s) \parallel p_\omega(s)) + \mathbb{E}_{d(s)}\left[\mathrm{KL}(\pi_\theta(a|s) \parallel \pi_\omega(a|s))\right]. \quad (24)$$

Observe that the first term in Eq. (24) does not depend on $\theta$, hence taking the minimum yields our desired result:

$$\max_\theta \mathcal{L}(\omega, \theta) = \min_\theta \left(\mathrm{KL}(d(s) \parallel p_\omega(s)) + \mathbb{E}_{d(s)}\left[\mathrm{KL}(\pi_\theta(a|s) \parallel \pi_\omega(a|s))\right]\right),$$

$$= \min_\theta \mathbb{E}_{d(s)}\left[\mathrm{KL}(\pi_\theta(a|s) \parallel \pi_\omega(a|s))\right].$$

Since $\mathrm{KL}(\pi_\theta(a|s) \parallel \pi_\omega(a|s)) \geq 0$, it follows that under exact representability, that is there exists $\theta \in \Theta$ s.t. $\pi_\theta(a|s) = \pi_\omega(a|s)$ and hence $\mathrm{KL}(\pi_\theta(a|s) \parallel \pi_\omega(a|s)) = 0$, we have $\min_\theta \mathbb{E}_{d(s)}\left[\mathrm{KL}(\pi_\theta(a|s) \parallel \pi_\omega(a|s))\right] = 0$. $\qquad\square$

# E Deriving the EM Algorithm

## E.1 E-Step

Here we provide a full derivation of our E-step of our variational actor-critic algorithm. The ELBO for our model from Eq. (4) with $\omega_k$ fixed is:

$$\mathcal{L}(\omega_k, \theta) = \mathbb{E}_{s\sim d(s)}\left[\frac{\mathbb{E}_{a\sim\pi_\theta(a|s)}\left[\hat{Q}_{\omega_k}(h)\right]}{\varepsilon_{\omega_k}} + \mathscr{H}(\pi_\theta(a|s))\right].$$

Taking derivatives of the $\omega$-fixed ELBO with respect to $\theta$ yields:

$$\nabla_\theta \mathcal{L}(\omega_k, \theta) = \mathbb{E}_{s\sim d(s)}\left[\frac{\nabla_\theta \mathbb{E}_{a\sim\pi_\theta(a|s)}\left[\hat{Q}_{\omega_k}(h)\right]}{\varepsilon_{\omega_k}} + \nabla_\theta \mathscr{H}(\pi_\theta(a|s))\right],$$

$$= \mathbb{E}_{s\sim d(s)}\left[\frac{\mathbb{E}_{a\sim\pi_\theta(a|s)}\left[\hat{Q}_{\omega_k}(h)\nabla_\theta \log \pi_\theta(a|s)\right]}{\varepsilon_{\omega_k}} + \nabla_\theta \mathscr{H}(\pi_\theta(a|s))\right],$$

where we have used the log-derivative trick [56] in deriving the final line. Note that in this form, when $\varepsilon_{\omega_k} \approx 0$, our gradient signal becomes very large. To prevent ill-conditioning, we multiply our objective by the constant $\varepsilon_{\omega_k}$. As $\varepsilon_{\omega_k} > 0$ for all non-optimal $\omega_k$ (see Theorem 2), this will not change the solution to the E-step optimisation. Our gradient becomes:

$$\varepsilon_{\omega_k} \nabla_\theta \mathcal{L}(\omega_k, \theta) = \mathbb{E}_{s \sim d(s)} \left[ \mathbb{E}_{a \sim \pi_\theta(a|s)} \left[ \hat{Q}_{\omega_k}(h) \nabla_\theta \log \pi_\theta(a|s) \right] + \varepsilon_{\omega_k} \nabla_\theta \mathscr{H}(\pi_\theta(a|s)) \right], \quad (25)$$

as required.

### E.2 M-Step

Here we provide a full derivation of the M-step for our variational actor-critic algorithm. The ELBO from Eq. (4) with $\theta_{k+1}$ fixed is:

$$\mathcal{L}(\omega, \theta_{k+1}) = \mathbb{E}_{d(s)} \left[ \frac{\mathbb{E}_{\pi_{\theta_{k+1}}(a|s)} \left[ \hat{Q}_\omega(h) \right]}{\varepsilon_\omega} + \mathscr{H}(\pi_{\theta_{k+1}}(a|s)) \right]$$

Taking derivatives of the with respect to $\omega$ yields:

$$\nabla_\omega \mathcal{L}(\omega, \theta_{k+1}) = \mathbb{E}_{d(s)\pi_{\theta_{k+1}}(a|s)} \left[ \nabla_\omega \left( \frac{\hat{Q}_\omega(h)}{\varepsilon_\omega} \right) \right],$$

$$= \mathbb{E}_{d(s)\pi_{\theta_{k+1}}(a|s)} \left[ \frac{\nabla_\omega \hat{Q}_\omega(h)}{\varepsilon_\omega} - \frac{\hat{Q}_\omega(h)}{(\varepsilon_\omega)^2} \nabla_\omega \varepsilon_\omega \right],$$

$$= \frac{1}{\varepsilon_\omega} \mathbb{E}_{d(s)\pi_{\theta_{k+1}}(a|s)} \left[ \nabla_\omega \hat{Q}_\omega(h) \right] - \frac{1}{(\varepsilon_\omega)^2} \mathbb{E}_{d(s)\pi_{\theta_{k+1}}(a|s)} \left[ \hat{Q}_\omega(h) \right] \nabla_\omega \varepsilon_\omega,$$

where we note that $\varepsilon_\omega$ does not depend on $h$, which allowed us to move it in and out of the expectation in deriving the final line. The gradient depends on terms up to $\frac{1}{(\varepsilon_\omega)^2}$, and so we multiply our objective by $(\varepsilon_{\omega_i})^2$ to prevent ill-conditioning when $\varepsilon_\omega \approx 0$. As $(\varepsilon_{\omega_i})^2 > 0$ for all non-convergent $\omega^*$, this does not change the solution to our M-step optimisation and can be seen as introducing an adaptive step size which supplements $\alpha_{\text{critic}}$. Observe that $\frac{\varepsilon_{\omega_i}}{\varepsilon_\omega} \big|_{\omega=\omega_i} = 1$, which, with a slight abuse of notation, yields our desired result:

$$(\varepsilon_{\omega_i})^2 \nabla_\omega \mathcal{L}(\omega, \theta_{k+1}) = \varepsilon_{\omega_i} \mathbb{E}_{d(s)\pi_{\theta_{k+1}}(a|s)} \left[ \nabla_\omega \hat{Q}_\omega(h) \right] - \mathbb{E}_{d(s)\pi_{\theta_{k+1}}(a|s)} \left[ \hat{Q}_\omega(h) \right] \nabla_\omega \varepsilon_\omega.$$

In general, calculating the exact gradient of $\varepsilon_\omega$ is non-trivial. We now derive this update for three important cases:

### E.3 Gradient of the Residual Error

We define $\beta_\omega(h) := \mathcal{T}_\omega \hat{Q}_\omega(h) - \hat{Q}_\omega(h)$ and use the notation $\mathbb{E}[\cdot] \triangleq \mathbb{E}_{h \sim \mathcal{U}(h)}[\cdot]$. Taking the derivative yields:

$$\nabla_\omega \varepsilon_\omega = \frac{1}{2|\mathcal{H}|} \nabla_\omega \|\beta_\omega(h)^2\|_2^2,$$

$$= \frac{1}{2} \nabla_\omega \mathbb{E} \left[ \beta_\omega(h)^2 \right],$$

$$= \mathbb{E} \left[ \beta_\omega(h) \nabla_\omega \beta_\omega(h) \right]. \quad (26)$$

For targets that do not depend on $\pi_\omega(a|s)$, the gradient of $\nabla_\omega \beta_\omega(h)$ can be computed directly. As an example, consider the update for the optimal Bellman operator target $\mathcal{T}^* \cdot := r(h) + \gamma \mathbb{E}_{h' \sim p(s'|h)} [\max_{a'}(\cdot)]$:

$$\nabla_\omega \beta_\omega(h) = \mathbb{E}_{s' \sim p(s'|h)} \left[ \nabla_\omega \hat{Q}_\omega(a^*, s') \right] - \nabla_\omega \hat{Q}_\omega(h),$$

where $a^* = \arg\max_a \hat{Q}(a, s')$.

Consider instead the Bellman operator target $\mathcal{T}^{\pi_\omega} \hat{Q}_\omega(h) = r(h) + \gamma \mathbb{E}_\omega \left[ \hat{Q}_\omega(h') \right]$ for the Bellman policy $\pi_\omega(a|s)$, which does have dependency on $\pi_\omega(a|s)$. For convenience, we denote the expectation $\mathbb{E}_{h' \sim p(s'|h)\pi_\omega(a'|s')} [\cdot]$ as $\mathbb{E}_\omega [\cdot]$. To obtain an analytic gradient, we must solve a recursive equation for $\nabla_\omega \pi_\omega(a|s)$. Consider the gradient of $\beta_\omega(h)$ with respect to $\omega$ using this operator:

$$
\begin{aligned}
\nabla_\omega \beta_\omega(h) &= \nabla_\omega \left( r(h) + \gamma \mathbb{E}_\omega \left[ \hat{Q}_\omega(h') \right] - \hat{Q}_\omega(h) \right), \\
&= \nabla_\omega \gamma \mathbb{E}_\omega \left[ \hat{Q}_\omega(h') \right] - \nabla_\omega \hat{Q}_\omega(h), \\
&= \gamma \mathbb{E}_\omega \left[ (\nabla_\omega \log \pi_\omega(a'|s')) \hat{Q}_\omega(h') + \nabla_\omega \hat{Q}_\omega(h') \right] - \nabla_\omega \hat{Q}_\omega(h), \\
&= \gamma \mathbb{E}_\omega \left[ (\nabla_\omega \log \pi_\omega(a'|s')) \hat{Q}_\omega(h') \right] + \gamma \mathbb{E}_\omega \left[ \nabla_\omega \hat{Q}_\omega(h') \right] - \nabla_\omega \hat{Q}_\omega(h), \\
&= \gamma \mathbb{E}_\omega \left[ (\nabla_\omega \log \pi_\omega(a'|s')) \hat{Q}_\omega(h') \right] + \Gamma_\omega(h),
\end{aligned}
\tag{27}
$$

where $\Gamma_\omega(h) := \gamma \mathbb{E}_\omega \left[ \nabla_\omega \hat{Q}_\omega(h') \right] - \nabla_\omega \hat{Q}_\omega(h)$. Substituting Eq. (27) into Eq. (26), we obtain:

$$
\begin{aligned}
\nabla_\omega \varepsilon_\omega &= \mathbb{E} \left[ \beta_\omega(h) \nabla_\omega \beta_\omega(h) \right], \\
&= \gamma \mathbb{E} \left[ \beta_\omega(h) \mathbb{E}_\omega \left[ (\nabla_\omega \log \pi_\omega(a'|s')) \hat{Q}_\omega(h') \right] \right] + \mathbb{E} \left[ \beta_\omega(h) \Gamma_\omega(h) \right]
\end{aligned}
\tag{28}
$$

To find an analytic expression for the first term of Eq. (28), we rely on the following theorem:
**Theorem 4** (Analytic Expression for Derivative of Boltzmann Policy Under Expectation). *If $\pi_\omega(a|s)$ is the Boltzmann policy defined in Eq. (3), it follows that:*

$$
\mathbb{E} \left[ \beta_\omega(h) \mathbb{E}_\omega \left[ (\nabla_\omega \log \pi_\omega(a'|s')) \hat{Q}_\omega(h') \right] \right] = \frac{\varepsilon_\omega \mathbb{E} \left[ \beta_\omega(h) \Gamma_\omega(h) \right] \mathcal{E}_\omega \hat{Q}_\omega(h) + \mathcal{E}_\omega \left[ \nabla_\omega \hat{Q}_\omega(h) \right]}{(\varepsilon_\omega)^2 \left( 1 + \gamma \mathbb{E} \left[ \beta_\omega(h) \mathbb{E}_\omega \left[ \hat{Q}_\omega(h') \right] \right] \right)},
$$

*where $\mathcal{E}_\omega$ is the operator $\mathcal{E}_\omega \cdot := \mathbb{E} \left[ \beta_\omega(h) \mathbb{E}_\omega \left[ \hat{Q}_\omega(h') \mathcal{M}_\omega \cdot \right] \right]$ and $\mathcal{M}_\omega$ denotes the operator $\mathcal{M}_\omega[\cdot] := \cdot - \mathbb{E}_{a \sim \pi_\omega(a|s)} [\cdot]$*

*Proof.* consider the derivative $\pi_\omega(a|s) \nabla_\omega \log \pi_\omega(a|s)$:

$$
\begin{aligned}
\pi_\omega(a|s) \nabla_\omega \log \pi_\omega(a|s) &= \nabla_\omega \pi_\omega(a|s), \\
&= \nabla_\omega \frac{\exp\left( \frac{\hat{Q}_\omega(h)}{\varepsilon_\omega} \right)}{\int_\mathcal{A} \exp\left( \frac{\hat{Q}_\omega(h)}{\varepsilon_\omega} \right) da}, \\
&= \nabla_\omega \left( \frac{\hat{Q}_\omega(h)}{\varepsilon_\omega} \right) \frac{\exp\left( \frac{\hat{Q}_\omega(h)}{\varepsilon_\omega} \right)}{\int_\mathcal{A} \exp\left( \frac{\hat{Q}_\omega(h)}{\varepsilon_\omega} \right) da} \\
&\quad - \frac{\exp\left( \frac{\hat{Q}_\omega(h)}{\varepsilon_\omega} \right)}{\int_\mathcal{A} \exp\left( \frac{\hat{Q}_\omega(h)}{\varepsilon_\omega} \right) da} \cdot \frac{\int_\mathcal{A} \nabla_\omega \left( \frac{\hat{Q}_\omega(h)}{\varepsilon_\omega} \right) \exp\left( \frac{\hat{Q}_\omega(h)}{\varepsilon_\omega} \right) da}{\int_\mathcal{A} \exp\left( \frac{\hat{Q}_\omega(h)}{\varepsilon_\omega} \right) da}, \\
&= \nabla_\omega \left( \frac{\hat{Q}_\omega(h)}{\varepsilon_\omega} \right) \pi_\omega(a|s) - \pi_\omega(a|s) \int_\mathcal{A} \nabla_\omega \left( \frac{\hat{Q}_\omega(h)}{\varepsilon_\omega} \right) \pi_\omega(a|s) da, \\
&= \nabla_\omega \left( \frac{\hat{Q}_\omega(h)}{\varepsilon_\omega} \right) \pi_\omega(a|s) - \pi_\omega(a|s) \mathbb{E}_{a \sim \pi_\omega(a|s)} \left[ \nabla_\omega \left( \frac{\hat{Q}_\omega(h)}{\varepsilon_\omega} \right) \right], \\
&= \pi_\omega(a|s) \left( \nabla_\omega \left( \frac{\hat{Q}_\omega(h)}{\varepsilon_\omega} \right) - \mathbb{E}_{a \sim \pi_\omega(a|s)} \left[ \nabla_\omega \left( \frac{\hat{Q}_\omega(h)}{\varepsilon_\omega} \right) \right] \right).
\end{aligned}
\tag{29}
$$

Finding an expression for $\nabla_\omega \left( \frac{\hat{Q}_\omega(h)}{\varepsilon_\omega} \right)$, we have:

$$\nabla_\omega \left( \frac{\hat{Q}_\omega(h)}{\varepsilon_\omega} \right) = \frac{1}{(\varepsilon_\omega)^2} \left( \varepsilon_\omega \nabla_\omega \hat{Q}_\omega(h) - \hat{Q}_\omega(h) \nabla_\omega \varepsilon_\omega \right).$$

Substituting into Eq. (29), we obtain:

$$
\begin{aligned}
\pi_\omega(a|s) \nabla_\omega \log \pi_\omega(a|s) &= \frac{\pi_\omega(a|s)}{(\varepsilon_\omega)^2} \Bigg( \varepsilon_\omega \nabla_\omega \hat{Q}_\omega(h) - \hat{Q}_\omega(h) \nabla_\omega \varepsilon_\omega \\
&\qquad - \mathbb{E}_{a \sim \pi_\omega(a|s)} \left[ \varepsilon_\omega \nabla_\omega \hat{Q}_\omega(h) - \hat{Q}_\omega(h) \nabla_\omega \varepsilon_\omega \right] \Bigg), \\
&= \frac{\pi_\omega(a|s)}{(\varepsilon_\omega)^2} \Bigg( \varepsilon_\omega \left( \nabla_\omega \hat{Q}_\omega(h) - \mathbb{E}_{a \sim \pi_\omega(a|s)} \left[ \nabla_\omega \hat{Q}_\omega(h) \right] \right) \\
&\qquad + \nabla_\omega \varepsilon_\omega \left( \mathbb{E}_{a \sim \pi_\omega(a|s)} \left[ \hat{Q}_\omega(h) \right] - \hat{Q}_\omega(h) \right) \Bigg), \\
&= \frac{\pi_\omega(a|s)}{(\varepsilon_\omega)^2} \left( \varepsilon_\omega \mathcal{M}_\omega \left[ \nabla_\omega \hat{Q}_\omega(h) \right] - \nabla_\omega \varepsilon_\omega \mathcal{M}_\omega \hat{Q}_\omega(h) \right),
\end{aligned}
$$

where $\mathcal{M}_\omega$ denotes the operator $\mathcal{M}_\omega[\cdot] := \cdot - \mathbb{E}_{a \sim \pi_\omega(a|s)}[\cdot]$. Dividing both sides by $\pi_\omega(a|s)$ yields:

$$\nabla_\omega \log \pi_\omega(a|s) = \frac{1}{(\varepsilon_\omega)^2} \left( \varepsilon_\omega \mathcal{M}_\omega \left[ \nabla_\omega \hat{Q}_\omega(h) \right] - \nabla_\omega \varepsilon_\omega \mathcal{M}_\omega \hat{Q}_\omega(h) \right).$$

Now, substituting for $\nabla_\omega \varepsilon_\omega = \mathbb{E}\left[ \beta_\omega(h) \nabla_\omega \beta_\omega(h) \right]$ from Eq. (26) yields:

$$\nabla_\omega \log \pi_\omega(a|s) = \frac{1}{(\varepsilon_\omega)^2} \left( \varepsilon_\omega \mathcal{M}_\omega \left[ \nabla_\omega \hat{Q}_\omega(h) \right] - \mathbb{E}\left[ \beta_\omega(h) \nabla_\omega \beta_\omega(h) \right] \mathcal{M}_\omega \hat{Q}_\omega(h) \right).$$

Now substituting for $\nabla_\omega \beta_\omega(h) = \gamma \mathbb{E}_\omega \left[ (\nabla_\omega \log \pi_\omega(a'|s')) \hat{Q}_\omega(h') \right] + \Gamma_\omega(h)$ from Eq. (27), and re-arranging for $\nabla_\omega \log \pi_\omega(a|s)$:

$$
\begin{aligned}
\nabla_\omega \log \pi_\omega(a|s) &= \frac{1}{(\varepsilon_\omega)^2} \Bigg( \varepsilon_\omega \mathcal{M}_\omega \left[ \nabla_\omega \hat{Q}_\omega(h) \right] - \gamma \mathbb{E}\left[ \beta_\omega(h) \mathbb{E}_\omega \left[ (\nabla_\omega \log \pi_\omega(a'|s')) \hat{Q}_\omega(h') \right] \right] \\
&\qquad\qquad + \mathbb{E}\left[ \beta_\omega(h) \Gamma_\omega(h) \right] \mathcal{M}_\omega \hat{Q}_\omega(h) \Bigg), \\
\nabla_\omega \log \pi_\omega(a|s) + \gamma \mathbb{E}\left[ \beta_\omega(h) \mathbb{E}_\omega \left[ (\nabla_\omega \log \pi_\omega(a'|s')) \hat{Q}_\omega(h') \right] \right] &= \frac{1}{(\varepsilon_\omega)^2} \Bigg( \varepsilon_\omega \mathcal{M}_\omega \left[ \nabla_\omega \hat{Q}_\omega(h) \right] \\
&\qquad\qquad + \mathbb{E}\left[ \beta_\omega(h) \Gamma_\omega(h) \right] \mathcal{M}_\omega \hat{Q}_\omega(h) \Bigg).
\end{aligned}
$$

Now, to obtain our desired result, we first multiply both sides by $\hat{Q}_\omega(h)$, take the expectation $\mathbb{E}_\omega$, multiply by $\beta_\omega(h)$ and finally take the expectation $\mathbb{E}$:

$$\mathbb{E}\left[\beta_\omega(h)\mathbb{E}_\omega\left[(\nabla_\omega \log \pi_\omega(a'|s'))\hat{Q}_\omega(h')\right]\right]\left(1 + \gamma\mathbb{E}\left[\beta_\omega(h)\mathbb{E}_\omega\left[\hat{Q}_\omega(h')\right]\right]\right)$$

$$= \frac{1}{(\varepsilon_\omega)^2}\left(\varepsilon_\omega\mathbb{E}\left[\beta_\omega(h)\mathbb{E}_\omega\left[\hat{Q}_\omega(h')\mathcal{M}_\omega\left[\nabla_\omega\hat{Q}_\omega(h')\right]\right]\right]\right.$$

$$\left. + \mathbb{E}\left[\beta_\omega(h)\Gamma_\omega(h)\right]\mathbb{E}\left[\beta_\omega(h)\mathbb{E}_\omega\left[\hat{Q}_\omega(h')\mathcal{M}_\omega\hat{Q}_\omega(h')\right]\right]\right).$$

$$\mathbb{E}\left[\beta_\omega(h)\mathbb{E}_\omega\left[(\nabla_\omega \log \pi_\omega(a'|s'))\hat{Q}_\omega(h')\right]\right] = \frac{\mathbb{E}\left[\beta_\omega(h)\mathbb{E}_\omega\left[\hat{Q}_\omega(h')\mathcal{M}_\omega\left[\nabla_\omega\hat{Q}_\omega(h')\right]\right]\right]}{\varepsilon_\omega\left(1 + \gamma\mathbb{E}\left[\beta_\omega(h)\mathbb{E}_\omega\left[\hat{Q}_\omega(h')\right]\right]\right)}$$

$$+ \frac{\mathbb{E}\left[\beta_\omega(h)\Gamma_\omega(h)\right]\mathbb{E}\left[\beta_\omega(h)\mathbb{E}_\omega\left[\hat{Q}_\omega(h')\mathcal{M}_\omega\hat{Q}_\omega(h')\right]\right]}{(\varepsilon_\omega)^2\left(1 + \gamma\mathbb{E}\left[\beta_\omega(h)\mathbb{E}_\omega\left[\hat{Q}_\omega(h')\right]\right]\right)},$$

$$= \frac{\varepsilon_\omega\mathbb{E}\left[\beta_\omega(h)\Gamma_\omega(h)\right]\mathcal{E}_\omega\hat{Q}_\omega(h) + \mathcal{E}_\omega\left[\nabla_\omega\hat{Q}_\omega(h)\right]}{(\varepsilon_\omega)^2\left(1 + \gamma\mathbb{E}\left[\beta_\omega(h)\mathbb{E}_\omega\left[\hat{Q}_\omega(h')\right]\right]\right)},$$

as required. $\qquad\square$

Using Theorem 4 to substitute for $\mathbb{E}\left[\beta_\omega(h)\mathbb{E}_\omega\left[(\nabla_\omega \log \pi_\omega(a'|s'))\hat{Q}_\omega(h')\right]\right]$ into Eq. (27), we obtain the result:

$$\nabla\varepsilon_\omega = \frac{\varepsilon_\omega\mathbb{E}\left[\beta_\omega(h)\Gamma_\omega(h)\right]\mathcal{E}_\omega\hat{Q}_\omega(h) + \mathcal{E}_\omega\left[\nabla_\omega\hat{Q}_\omega(h)\right]}{(\varepsilon_\omega)^2\left(1 + \gamma\mathbb{E}\left[\beta_\omega(h)\mathbb{E}_\omega\left[\hat{Q}_\omega(h')\right]\right]\right)} + \mathbb{E}\left[\beta_\omega(h)\Gamma_\omega(h)\right]. \tag{30}$$

The second term of Eq. (30) is the standard policy evaluation gradient and the first term changes $\pi_\omega(a|s)$ in the direction of increasing $\varepsilon_\omega$. We see that all expectations in Eq. (30) can be approximated by sampling from our variational policy $\pi_\theta(a|s) \approx \pi_\omega(a|s)$. After a complete E-step, and under Assumption 4, we have $\pi_\theta(a|s) = \pi_\omega(a|s)$ and the gradient is exact.

While the first term in Eq. (30) is certainly tractable, it presents a formidable challenge for the programmer to implement, especially if unbiased estimates are required; several expressions which involve the multiplication of more than one expectation $\mathbb{E}_\omega$ need to be evaluated. In all of these cases, expectations approximated using the same data will introduce bias, however it is often infeasible to sample more than once from the same state in the environment. Like in Sutton et al. [58], a solution to this problem is to learn a function approximator for one of the expectations that is updated at a slower rate than the other expectation. Alternatively, these function approximators can be updated using separate data batches from a replay buffer.

A radical approach is to simply neglect this gradient term, which we discuss in Appendix F.3. A more considered approach is to use an operator that does not constraint $\Omega$. Consider the operator introduced in Appendix F.2,

$$\mathcal{T}_{\omega,k}\cdot = r(h) + \gamma\mathbb{E}_{\omega,k}[\cdot],$$

where we have used the shorthand for expectation $\mathbb{E}_{\omega,k}[\cdot] := \mathbb{E}_{h'\sim p(s'|h)p_{\omega,k}(a'|s')}[\cdot]$ and the Boltzmann distribution is defined as

$$p_{\omega,k}(a|s) := \frac{\exp\left(\frac{\hat{Q}_\omega(h)}{\varepsilon_k}\right)}{\int_\mathcal{A}\exp\left(\frac{\hat{Q}_\omega(h)}{\varepsilon_k}\right)da}.$$

The incremental residual error is defined as $\varepsilon_{\omega,k} := \frac{1}{2|\mathcal{H}|}\|\beta_{\omega,k}(h)\|_2^2 + \varepsilon_k$ and $\beta_{\omega,k}(h) := \mathcal{T}_{\omega,k}\hat{Q}_\omega(h) - \hat{Q}_\omega(h)$. Taking gradients of $\varepsilon_{\omega,k}$ directly yields:

$$\nabla_\omega\varepsilon_{\omega,k} = \mathbb{E}\left[\beta_{\omega,k}(h)\nabla_\omega\beta_{\omega,k}(h)\right].$$

where

$$\nabla_\omega \beta_{\omega,k}(h) = \nabla_\omega \mathbb{E}_{\omega,k}\left[\hat{Q}_\omega(h')\right] - \nabla_\omega \hat{Q}_\omega(h),$$

$$= \nabla_\omega \mathbb{E}_{\omega,k}\left[\hat{Q}_\omega(h')\right] - \nabla_\omega \hat{Q}_\omega(h),$$

$$= \mathbb{E}_{\omega,k}\left[\nabla_\omega \log p_{\omega,k}(a'|s') + \nabla_\omega \hat{Q}_\omega(h')\right] - \nabla_\omega \hat{Q}_\omega(h). \qquad (31)$$

Now, $\nabla_\omega \log p_{\omega,k}(a'|s')$ can be computed directly as:

$$\nabla_\omega \log p_{\omega,k}(a'|s') = \nabla_\omega \left( \frac{\hat{Q}_\omega(h')}{\varepsilon_k} - \log \int_{\mathcal{A}} \exp\left(\frac{\hat{Q}_\omega(h')}{\varepsilon_k}\right) da \right),$$

$$= \frac{\nabla_\omega \hat{Q}_\omega(h')}{\varepsilon_k} - \int_{\mathcal{A}} \frac{\nabla_\omega \hat{Q}_\omega(h')}{\varepsilon_k} \frac{\exp\left(\frac{\hat{Q}_\omega(h')}{\varepsilon_k}\right)}{\int_{\mathcal{A}} \exp\left(\frac{\hat{Q}_\omega(h)}{\varepsilon_k}\right) da} da,$$

$$= \frac{\nabla_\omega \hat{Q}_\omega(h')}{\varepsilon_k} - \int_{\mathcal{A}} \frac{\nabla_\omega \hat{Q}_\omega(h)}{\varepsilon_k} p_{\omega,k}(a'|s') da,$$

$$= \frac{\nabla_\omega \hat{Q}_\omega(h')}{\varepsilon_k} - \mathbb{E}_{a' \sim p_{\omega,k}(a'|s')}\left[\frac{\nabla_\omega \hat{Q}_\omega(h)}{\varepsilon_k}\right],$$

$$= \mathcal{M}_{\omega,k}\left[\frac{\nabla_\omega \hat{Q}_\omega(h')}{\varepsilon_k}\right],$$

where where $\mathcal{M}_{\omega,k}$ denotes the operator $\mathcal{M}_{\omega,k}[\cdot] := \cdot - \mathbb{E}_{a \sim p_{\omega,k}(a|s)}[\cdot]$. Substituting into Eq. (31) yields:

$$\nabla_\omega \beta_{\omega,k}(h) = \mathbb{E}_{\omega,k}\left[\mathcal{M}_{\omega,k}\left[\frac{\nabla_\omega \hat{Q}_\omega(h')}{\varepsilon_k}\right] + \nabla_\omega \hat{Q}_\omega(h')\right] - \nabla_\omega \hat{Q}_\omega(h).$$

### E.4 Discussion of E-step

We now explore the relationship between classical actor-critic methods and the E-step. The policy gradient theorem [56] derives an update for the derivative of the RL objective (1) with respect to the policy parameters

$$\nabla_\theta J(\theta) = \mathbb{E}_{s \sim \rho^\pi(s)}\left[\mathbb{E}_{a \sim \pi_\theta(a|s)}\left[Q^\pi(h) \nabla_\theta \log \pi_\theta(a|s)\right]\right],$$

where $\rho^\pi(s)$ is the discounted-ergodic occupancy, defined formally in Ciosek & Whiteson [11], and in general not a normalised distribution. To obtain practical algorithms, we collect rollouts and treat them as samples from the steady-state distribution instead.

By contrast, the VIREL policy update in Eq. (25) involves an expectation over $d(s)$, which can be any sampling distribution decorrelated from $\pi$ ensuring all states are visited infinitely often. As $\hat{Q}_\omega(h)$ is also independent of $\pi_\theta(a|s)$, we can move the gradient operator $\nabla_\theta$ out of the inner integral to obtain

$$\mathbb{E}_{s \sim d(s)}\left[\mathbb{E}_{a \sim \pi_\theta(a|s)}\left[\hat{Q}_\omega(h) \nabla_\theta \log \pi_\theta(a|s)\right]\right] = \mathbb{E}_{s \sim d(s)}\left[\nabla_\theta \mathbb{E}_{a \sim \pi_\theta(a|s)}\left[\hat{Q}_\omega(h)\right]\right]$$

This transformation is essential in deriving powerful policy gradient methods such as Expected and Fourier Policy Gradients [10, 15] and holds for deterministic polices [53]. However, unlike in VIREL, it is not strictly justified in the classic policy gradient theorem [56] and MERL formulation [25].

## F  Relaxations and Approximations

### F.1  Relaxation of Representability of $Q$-functions

In our analysis, Assumption 2 is required by Theorem 2 to ensure that a maximum to the optimisation problem exists, however it can be completely neglected provided that projected Bellman operators

are used; moreover, if projected Bellman operators are used, our M-step is also always guaranteed to converge, even if our E-step does not. Consequently, we can terminate the algorithm by carrying out a complete M-step at any time using our variational approximation and still be guaranteed convergence to a sub-optimal point.

We now introduce the assumption that our action-value function approximator is three-times differentiable over $\Omega$, which is required for convergence guarantees.

**Assumption 5** (Universal Smoothness of $\hat{Q}_\omega(h)$). *We require that $\hat{Q}_\omega(h) \in \mathbb{C}^3(\Omega)$ for all $h \in \mathcal{H}$,*

We now extend the analysis of Bhatnagar et al. [6] to continuous domains. Consider the local linearisation of the function approximator $\hat{Q}_\omega(h) \approx b_\omega^\top(h)\omega$, where $b_\omega(h) := \nabla_\omega \hat{Q}_\omega(h)$. We define the projection operator $\mathcal{P}_\omega Q(\cdot) := b_\omega^\top(h)\omega'$ where $\tilde{\omega}$ are the parameters that minimise the difference between the action-value function and the local linearisation:

$$\tilde{\omega} := \arg\min_{\omega'} \frac{1}{2|\mathcal{H}|} \|Q(h) - b_\omega^\top(h)\omega'\|_2^2. \tag{32}$$

Using the notation $\mathbb{E}[\cdot] \triangleq \mathbb{E}_{h \sim \mathcal{U}(h)}[\cdot]$ and taking derivatives of Eq. (32) with respect to $\omega'$ yields:

$$\nabla_{\omega'} \frac{1}{2|\mathcal{H}|} \|Q(h) - {\omega'}^\top\|_2^2 = \frac{1}{2}\nabla_{\omega'}\mathbb{E}\left[(Q(h) - b_\omega^\top(h)\omega')^2\right],$$

$$= \frac{1}{2}\mathbb{E}\left[\nabla_{\omega'}(Q(h)^2 - 2b_\omega^\top(h)\omega' Q(h) + b_\omega^\top(h)\omega' b_\omega^\top(h)\omega')\right],$$

$$= \mathbb{E}\left[b_\omega(h)b_\omega^\top(h)\omega' - b_\omega(h)Q(h)\right].$$

Equating to zero and solving for $\tilde{\omega}$, we obtain:

$$\tilde{\omega} = \mathbb{E}\left[b_\omega(h)b_\omega^\top(h)\right]^{-1} \mathbb{E}\left[b_\omega(h)Q(h)\right].$$

Substituting into our operator yields:

$$\mathcal{P}_\omega \cdot = b_\omega^\top(h)\mathbb{E}\left[b_\omega(h)b_\omega^\top(h)\right]^{-1} \mathbb{E}\left[b_\omega(h)\cdot\right].$$

We can therefore interpret $\mathcal{P}$ as an operator that projects an action-value function onto the tangent space of $\hat{Q}_\omega(h)$ at $\omega$. For linear function approximators of the form $\hat{Q}_\omega(h) = b^\top(h)\omega$, the projection operator is independent of $\omega$ and projects $Q$ directly onto the nearest function approximator and the operator [57].

We now replace the residual error in Section 3.1 with the projected residual error,

$$\varepsilon_\omega := \frac{1}{2|\mathcal{H}|} \left\|\mathcal{P}_\omega\left(\mathcal{T}_\omega \hat{Q}_\omega(h) - \hat{Q}_\omega(h)\right)\right\|_2^2. \tag{33}$$

By definition, there always exists fixed point $\omega \in \Omega$ for which $\varepsilon_\omega = 0$, which means that $\varepsilon_\omega$ now satisfies all requirements in Theorem 2 without Assumption 2. We can also carry out a complete partial variational M-step by minimising the surrogate $\varepsilon_\omega$, keeping $\pi_\omega(a|s) = \pi_\theta(a|s)$ in all expectations. At convergence, we have $\varepsilon_\omega = 0$ in this case.

We now derive the more convenient form of $\varepsilon_\omega$ from Lemma 1 in Bhatnagar et al. [6], extending this result to continuous domains. Let $\beta_\omega(h) := \mathcal{T}_\omega \hat{Q}_\omega(h) - \hat{Q}_\omega(h)$. Substituting into Eq. (33), we obtain:

$$2\varepsilon_\omega = \frac{1}{|\mathcal{H}|} \|\mathcal{P}_\omega \beta_\omega(h)\|_2^2,$$

$$= \frac{1}{|\mathcal{H}|} \left\|b_\omega^\top(h)\mathbb{E}\left[b_\omega(h)b_\omega^\top(h)\right]^{-1} \mathbb{E}\left[b_\omega(h)\beta_\omega(h)\right]\right\|_2^2,$$

$$= \mathbb{E}\left[\mathbb{E}\left[b_\omega^\top(h)\beta_\omega(h)\right] \mathbb{E}\left[b_\omega(h)b_\omega^\top(h)\right]^{-1} b_\omega(h)b_\omega^\top(h)\mathbb{E}\left[b_\omega(h)b_\omega^\top(h)\right]^{-1} \mathbb{E}\left[\beta_\omega(h)b_\omega(h)\right]\right],$$

$$= \mathbb{E}\left[b_\omega^\top(h)\beta_\omega(h)\right] \mathbb{E}\left[b_\omega(h)b_\omega^\top(h)\right]^{-1} \mathbb{E}\left[b_\omega(h)b_\omega^\top(h)\right] \mathbb{E}\left[b_\omega(h)b_\omega^\top(h)\right]^{-1} \mathbb{E}\left[\beta_\omega(h)b_\omega(h)\right],$$

$$= \mathbb{E}\left[b_\omega^\top(h)\beta_\omega(h)\right] \mathbb{E}\left[b_\omega(h)b_\omega^\top(h)\right]^{-1} \mathbb{E}\left[\beta_\omega(h)b_\omega(h)\right].$$

Denoting $\zeta_\omega := \mathbb{E}\left[b_\omega(h)b_\omega^\top(h)\right]^{-1}\mathbb{E}\left[\beta_\omega(h)b_\omega(h)\right]$ following the analysis in [6], we find the derivative of $\varepsilon_\omega$ as:

$$\nabla_\omega\varepsilon_\omega = \mathbb{E}\left[(\nabla_\omega\beta_\omega(h))b_\omega^\top(h)\zeta_\omega\right] + \mathbb{E}\left[(\beta_\omega(h) - b_\omega^\top(h)\zeta_\omega)\nabla_\omega^2\hat{Q}_\omega(h)\zeta_\omega\right].$$

Following the method of Pearlmutter [45], the multiplication between the Hessian and $\zeta_\omega$ can be calculated in $O(n)$ time, which bounds the overall complexity of our algorithm. To avoid bias in our estimate, we learn a set of weights $\hat{\zeta} \approx \zeta_\omega$ on a slower timescale, which we update as:

$$\hat{\zeta}_{k+1} \leftarrow \hat{\zeta}_k + \alpha_{\zeta k}\left(\beta_\omega(h) - b_\omega^\top(h)\zeta_k\right)b_\omega(h), \tag{34}$$

where $\alpha_{\zeta k}$ is a step size chosen to ensure that $\alpha_{\zeta k} < \alpha_{\text{critic}}$. The weights are then used to find our gradient term:

$$\nabla_\omega\varepsilon_\omega = \mathbb{E}\left[(\nabla_\omega\beta_\omega(h))b_\omega^\top(h)\hat{\zeta}\right] + \mathbb{E}\left[(\beta_\omega(h) - b_\omega^\top(h)\hat{\zeta})\nabla_\omega^2\hat{Q}_\omega(h)\zeta_\omega\right].$$

In our framework, the term $\nabla\beta_\omega(h)$ is specific to our choice of operator. In Bhatnagar et al. [6], a TD-target is used and parameter updates for $\omega$ are given as:

$$\omega_{k+1} = \mathfrak{P}\left(\omega_k + \alpha_{\omega k}(b_k - \gamma b_k')b_k^\top\hat{\zeta}_k - q_k\right), \tag{35}$$

$$q_k := \left(\beta_{\omega_k}(h_k) - b_k^\top\hat{\zeta}_k\right)\nabla_\omega^2\hat{Q}_{\omega_k}(h_k)\hat{\zeta}_k$$

where $b_k := b_{\omega_k}(h_k)$ and $\mathfrak{P}(\cdot)$ is an operator that projects $\omega_k$ into any arbitrary compact set with a smooth boundary, $\mathcal{C}$. The projection $\mathfrak{P}(\cdot)$ is introduced for mathematical formalism and, provided $\mathcal{C}$ is large enough to contain all solutions $\left\{\omega|\mathbb{E}\left[\beta_\omega(h)\nabla_\omega\hat{Q}_\omega(h)\right] = 0\right\} \subseteq \mathcal{C}$, has no bearing on the updates in practice. Under Assumption 5, provided the step size conditions $\sum_k^\infty \alpha_{\zeta k} = \sum_k^\infty \alpha_{\omega k} = \infty$, $\sum_k^\infty \alpha_{\zeta k}^2 <, \sum_k^\infty \alpha_{\omega k}^2 < \infty$ and $\lim_{k\to\infty}\frac{\alpha_{\zeta k}}{\alpha_{\omega k}} = 0$ hold and $\mathbb{E}[b_\omega(h)b_\omega^\top(h)]$ is non-singular $\forall\omega \in \Omega$, the analysis in Theorem 2 of Bhatnagar et al. [6] applies and the updates in Eqs. (34) and (35) are guaranteed to converge to the TD fixed point. This demonstrates using data sampled from any variational policy $\pi_\theta(a|s)$ to update $\omega_k$ as Eqs. (34) and (35), $\omega_k$ will converge to a fixed point.

### F.2 Off-Policy Bellman Operators

As discussed in Section 3.1, using the Bellman operator $\mathcal{T}^{\pi_\omega}\cdot$ induces a constraint on the set of parameters $\Omega$. While this constraint can be avoided using the optimal Bellman operator $\mathcal{T}^*\cdot := r(h) + \gamma\mathbb{E}_{h'\sim p(s'|h)}[\max_{a'}(\cdot)]$, evaluating $\max_{a'}(\hat{Q}_\omega(h'))$ may be difficult in large continuous domains. We now make a slight modification to our model in Section 3.1 to accommodate a Bellman operator that avoids these two practical difficulties.

Firstly, we introduce a new Boltzmann distribution $p_{\omega,k}(a|s)$:

$$p_{\omega,k}(a|s) := \frac{\exp\left(\frac{\hat{Q}_\omega(h)}{\varepsilon_k}\right)}{\int_\mathcal{A}\exp\left(\frac{\hat{Q}_\omega(h)}{\varepsilon_k}\right)da},$$

where $\{\varepsilon_k\}$ is a sequence of positive constants $\varepsilon_k \geq 0$, $\lim_{k\to\infty}\varepsilon_k = 0$. We now introduce a new operator $\mathcal{T}_{\omega,k}\cdot$, defined as is the Bellman operator for $p_{\omega,k}(a|s)$:

$$\mathcal{T}_{\omega,k}\cdot := \mathcal{T}^{p_{\omega,k}}\cdot = r(h) + \gamma\mathbb{E}_{h'\sim p(s'|h)p_{\omega,k}(a'|s')}[\cdot]. \tag{36}$$

Let $\pi_{\omega,k}(a|s)$ be the Boltzmann policy:

$$\pi_{\omega,k}(a|s) := \frac{\exp\left(\frac{\hat{Q}_\omega(h)}{\varepsilon_{\omega,k}}\right)}{\int_\mathcal{A}\exp\left(\frac{\hat{Q}_\omega(h)}{\varepsilon_{\omega,k}}\right)da},$$

where the residual error $\varepsilon_{\omega,k} := \frac{c}{p}\|\mathcal{T}_{\omega,k}\hat{Q}_\omega(h) - \hat{Q}_\omega(h)\|_p^p + \varepsilon_k$. It is clear that $\mathcal{T}_{\omega,k}\cdot$ does not constrain $\Omega$ as $\varepsilon_k$ has no dependency on $\omega$ and $\pi_{\omega,k}(a|s)$ is well defined for all $\omega \in \Omega$.

We now formally prove that $\min_\omega\lim_{k\to\infty}\varepsilon_{\omega,k} = \min_\omega\varepsilon_\omega$, and so minimising $\varepsilon_{\omega,k}$ is the same as minimising the objective $\varepsilon_\omega$ from Section 3.1 and that $\mathcal{T}_{\omega,k}\cdot \in \mathbb{T}$. We also prove that $\min_\omega\lim_{k\to\infty}\varepsilon_{\omega,k} = \lim_{k\to\infty}\min_\omega\varepsilon_{\omega,k}$ (i.e. that min and lim commute), which allows us to minimise our objective incrementally over sequences $\varepsilon_{\omega,k}$.

**Theorem 5** (Incremental Optimisation of $\varepsilon_{\omega,k}$). *Let $\varepsilon_{\omega,k} := \frac{c}{p}\|\mathcal{T}_{\omega,k}\hat{Q}_\omega(h) - \hat{Q}_\omega(h)\|_p^p + \varepsilon_k$ and $\mathcal{T}_{\omega,k}$ be the Bellman operator defined in Eq.* (36)*. It follows that i)* $\mathcal{T}_{\omega,k}\cdot \in \mathbb{T}$*, ii)* $\min_\omega \lim_{k\to\infty}\varepsilon_{\omega,k} = \min_\omega \varepsilon_\omega$ *and iii)* $\min_\omega \lim_{k\to\infty}\varepsilon_{\omega,k} = \lim_{k\to\infty}\min_\omega \varepsilon_{\omega,k}$

*Proof.* To prove i), we take the limit $\lim_{k\to\infty}\mathcal{T}_{\omega,k}\hat{Q}_\omega(h) = \mathcal{T}^*\hat{Q}_\omega(h)$:

$$\lim_{k\to\infty}\mathcal{T}_{\omega,k}\hat{Q}_\omega(h) = r(h) + \lim_{k\to\infty}\gamma\mathbb{E}_{h'\sim p(s'|h)p_{\omega,k}(a'|s')}\left[\hat{Q}_\omega(h)\right].$$

Observe that from Theorem 1, we have

$$\lim_{\varepsilon_k\to\infty}\gamma\mathbb{E}_{h'\sim p(s'|h)p_{\omega,k}(a'|s')}\left[\hat{Q}_\omega(h)\right] = \gamma\mathbb{E}_{h'\sim p(s'|h)\delta(a=\arg\max_{a'}(\hat{Q}_\omega(a',s)))}\left[\hat{Q}_\omega(h)\right],$$

hence:

$$\lim_{k\to\infty}\mathcal{T}_{\omega,k}\hat{Q}_\omega(h) = r(h) + \lim_{k\to\infty}\gamma\mathbb{E}_{h'\sim p(s'|h)p_{\omega,k}(a'|s')}\left[\hat{Q}_\omega(h)\right],$$

$$= r(h) + \gamma\mathbb{E}_{h'\sim p(s'|h)\delta(a=\arg\max_{a'}(\hat{Q}_\omega(a',s)))}\left[\hat{Q}_\omega(h)\right],$$

$$= r(h) + \gamma\mathbb{E}_{s'\sim p(s'|h)}\left[\max_{a'}(\hat{Q}_\omega(h))\right],$$

$$= \mathcal{T}^*\hat{Q}_\omega(h).$$

Our operator is therefore constructed such that in the limit $k\to\infty$, we recover the optimal Bellman operator. Observe too that as $\frac{c}{p}\|\mathcal{T}_{\omega,k}\hat{Q}_\omega(h) - \hat{Q}_\omega(h)\|_p^p \geq 0$, we have $\varepsilon_{\omega,k} > 0$ for all $\varepsilon_k > 0$. From Theorem 1, it follows that $\pi_{\omega,k}(a|s) \to \delta(a = \arg\max_{a'}(\hat{Q}_\omega(a', s)))$ only in the limit $\lim_{k\to\infty}\varepsilon_k = 0$ and when $\varepsilon_{\omega,k} = 0$. Under this limit, we have $\lim_{k\to\infty}\mathcal{T}_{\omega,k} = \mathcal{T}^*$ and so $\mathcal{T}_{\omega,k} \in \mathbb{T}$, as required for i).

To prove ii), consider taking the limit of $\varepsilon_{\omega,k}$ directly:

$$\lim_{k\to\infty}\varepsilon_{\omega,k} = \lim_{k\to\infty}\left(\frac{c}{p}\|\mathcal{T}_{\omega,k}\hat{Q}_\omega(h) - \hat{Q}_\omega(h)\|_p^p + \varepsilon_k\right),$$

$$= \lim_{k\to\infty}\left(\frac{c}{p}\|\mathcal{T}_{\omega,k}\hat{Q}_\omega(h) - \hat{Q}_\omega(h)\|_p^p\right) + \varepsilon_\infty,$$

$$= \frac{c}{p}\|\lim_{k\to\infty}\mathcal{T}_{\omega,k}\hat{Q}_\omega(h) - \hat{Q}_\omega(h)\|_p^p,$$

$$= \frac{c}{p}\|\mathcal{T}^*\hat{Q}_\omega(h) - \hat{Q}_\omega(h)\|_p^p,$$

$$= \varepsilon_\omega, \tag{37}$$

as required.

To prove iii), let $\tilde{\omega}_k$ be the minimiser of $\varepsilon_{\omega,k}$, that is $\tilde{\omega}_k = \arg\min_\omega \varepsilon_{\omega,k}$. Let $\tilde{\omega}$ be the limit of all such sequences $\tilde{\omega} = \lim_{k\to\infty}\tilde{\omega}_k$ and let $\omega^* = \arg\min_\omega \varepsilon_\omega$. By definition, we have $\varepsilon_{\tilde{\omega}_k,k} \leq \varepsilon_{\omega,k}$. Taking the limit $k\to\infty$ and then the $\min$, we have:

$$\min\lim_{k\to\infty}\varepsilon_{\tilde{\omega}_k,k} \leq \min\lim_{k\to\infty}\varepsilon_{\omega,k},$$

$$\implies \varepsilon_{\tilde{\omega},\infty} \leq \min\lim_{k\to\infty}\varepsilon_{\omega,k}. \tag{38}$$

Using Assumption 2 and Eq. (37), it follows that the right hand side of Eq. (38) is $\min\lim_{k\to\infty}\varepsilon_{\omega,k} = \min\varepsilon_\omega = 0$, hence $\varepsilon_{\tilde{\omega},\infty} \leq 0$. By definition, $\varepsilon_{\tilde{\omega},\infty} \geq 0$, and so equality must hold. It therefore follows $\lim_{k\to\infty}\min_\omega \varepsilon_{\omega,k} = \varepsilon_{\tilde{\omega},\infty} = 0$, which implies $\min_\omega \lim_{k\to\infty}\varepsilon_{\omega,k} = \lim_{k\to\infty}\min_\omega \varepsilon_\omega = 0$ as required. $\square$

Overall, this result permits us to carry out separate optimisations over $\varepsilon_{\omega,k}$ while gradually increasing $k\to\infty$ to obtain the same result as minimising $\varepsilon_\omega$ directly. The advantage to this method is that each minimisation $\varepsilon_{\omega,k}$ involves the operator $\mathcal{T}_{\omega,k}$, which is tractable, mathematically convenient and does not constrain $\Omega$. Note too that, as calculated in Appendix E.3, the gradient $\nabla_\omega\varepsilon_{\omega,k}$ is straightforward to implement in comparison with $\nabla_\omega\varepsilon_\omega$ using $\mathcal{T}^{\pi_\omega}$. We save investigating this operator further for future work.

## F.3 Approximate Gradient Methods and Partial Optimisation

A common trick in policy evaluation is to use a direct method [2, 55]. Like in supervised methods [7], direct methods treat the term $\mathcal{T}_\omega \hat{Q}_\omega(h)$ as a fixed target, rather than a differential function. Introducing the notation $\mathbb{E}[\cdot] \triangleq \mathbb{E}_{h \sim \mathcal{U}(h)}[\cdot]$, the gradient can easily be derived as:

$$\nabla_\omega \varepsilon_\omega = \frac{1}{2}\nabla_\omega \mathbb{E}\left[\left(\dashv\left[\mathcal{T}_\omega \hat{Q}_\omega(h)\right] - \hat{Q}_\omega(h)\right)^2\right],$$

$$= -\mathbb{E}\left[\left(\hat{Q}_\omega(h) - \mathcal{T}_\omega \hat{Q}_\omega(h)\right)\nabla_\omega \hat{Q}_\omega(h)\right]$$

where $\dashv[\cdot]$ is the stopgrad operator, which sets the gradient of its operand to zero, $\dashv[\cdot] = \cdot$, $\nabla \dashv[\cdot] = 0$ [16]. For general function approximators, direct methods have no convergence guarantees, and indeed there exist several famous examples of divergence when used with classic RL targets [5, 65, 70], however its ubiquity in the RL community is testament to its ease of implementation and empirical success [42, 55]. We therefore see no reason why it should not be successful for VIREL, a claim which we verify in Section 5. In our setting, we replace our M-step with the simplified objective $\omega_{k+1} \leftarrow \arg\min_\omega \varepsilon_\omega$. This is justified because $\arg\min_\omega \varepsilon_\omega$ was the original objective motivated in Section 3.1, and so the only limitation to minimising this directly is obtaining a good enough variational policy $\pi_\omega(a|s) \approx \pi_\theta(a|s)$. More formally, our objective $\mathcal{L}(\omega, \theta)$ is maximised for any $\varepsilon_\omega \to 0$, so $\arg\min_\omega \varepsilon_\omega$ can be considered a surrogate objective for $\mathcal{L}(\omega, \theta)$. Using direct methods, M-step update becomes:

**M-Step (Critic) direct:** $\quad \omega_{i+1} \leftarrow \omega_i - \alpha_{\text{critic}}\nabla_\omega \varepsilon_\omega|_{\omega=\omega_i}$,

$$\nabla_\omega \varepsilon_\omega = \mathbb{E}\left[\left(\hat{Q}_\omega(h) - \mathcal{T}_\omega \hat{Q}_\omega(h)\right)\nabla_\omega \hat{Q}_\omega(h)\right].$$

We can approximate $\mathcal{T}_\omega \hat{Q}_\omega(h)$ by sampling from the variational distribution $\pi_\theta(a|s)$ and by using any appropriate RL target. Another important approximation that we make is that we perform only partial E- and M-steps, halting optimisation before convergence. From a practical perspective, convergence can often only occur in a limit of infinite time steps anyway, and if good empirical performance result from taking partial E- and M-steps, computation may be wasted carrying out many sub-optimisation steps for little gain.

As analysed by Gunawardana & Byrne [23], such algorithms fall under the umbrella of the generalised alternating maximisation (GAM) framework, and convergence guarantees are specific to the form of function approximator and MDP. Like in many inference settings, we anticipate that most function approximators and MDPs of interest will not satisfy the conditions required to prove convergence, however variational EM procedures are known to be to empirically successful even when convergence properties are not guaranteed [23, 66]. We demonstrate in Section 5 that taking partial EM steps does not hinder our performance.

## F.4 Local Smoothness of $\hat{Q}_{\omega^*}(\cdot)$

For Theorem 1 to hold, we require that $\hat{Q}_{\omega^*}(\cdot)$ is locally smooth about its maximum. Our choice of function approximator may prevent this condition from holding, for example, a neural network with ReLU elements can introduce a discontinuity in gradient at $\max_h \hat{Q}_{\omega^*}(h)$. In practice, a formal Dirac-delta function can only ever emerge in the limit of convergence $\varepsilon_\omega \to 0$. In finite time, we obtain, at best, a nascent delta function; that is a function with very small variance that is 'on the way to convergence' (see, for example, Kelly [31] for a formal definition). The mode of a nascent delta function therefore approximates the true Dirac-delta distribution. When $\hat{Q}_{\omega^*}(\cdot)$ is not locally smooth, functions that behave similarly to nascent delta functions will still emerge at finite time, the mode of which we anticipate provides an approximation to the hardmax behaviour we require for most RL settings.

We also require that $\hat{Q}_{\omega^*}(\cdot)$ has a single, unique global maximum for any state. In reality, optimal Q-functions may have more than one global maxima for a single state corresponding to the existence of multiple optimal policies. To ensure Assumption 3 strictly holds, we can arbitrarily reduce the reward for all but one optimal policy. We anticipate that this is unnecessary in practice, as our

risk-neutral objective means that a variational policy will be encouraged fit to a single mode anyway. In addition, these assumptions are required to characterise behaviour under convergence to a solution and will not present a problem in finite time where $\hat{Q}_\omega(h)$ is very unlikely to have more than one global optimum.

## F.5 Analysis of Approximate EM Algorithms

We now provide two separate analyses of our EM algorithm, replacing the Bellman operator $\mathcal{T}^{\pi_\omega}\cdot$ with its unconstrained variational approximation $\mathcal{T}^{\pi_\theta}\cdot$ (effectively substituting for $\pi_\omega(\cdot|s) \approx \pi_\theta(\cdot|s)$ under expectation). In our first analysis, we make no simplifying assumptions on $\varepsilon_\omega$, showing that our EM algorithm reduces exactly to policy iteration and evaluation. In our second analysis, we use a direct method, treating $\mathcal{T}^{\pi_\theta}\cdot$ as a fixed target as outlined in Appendix F.3, showing that the algorithm reduces exactly to Q-learning.

In both analyses, we assume a complete E- and M- step can be carried out and our class of function approximators is rich enough to represent any action-value function. Let $\pi_{\theta_0}(a|\cdot)$ be any initial policy and $\hat{Q}_{\omega_0}(\cdot)$ an arbitrary initialisation of the function approximator. For notational convenience we write $\pi_k(a|\cdot) := \pi_{\theta_k}(a|\cdot)$.

**Analysis with $\omega$-Dependent Target**   As we prove in Theorem 2, we can always maximise our objective with respect to $\omega$ by finding $\omega^*$ s.t. $\varepsilon_{\omega^*} = 0$. This gives the M-step update:

$$\omega_1 = \arg\min_\omega \varepsilon_\omega,$$
$$\implies \varepsilon_{\omega_1} = 0,$$
$$\implies \mathcal{T}^{\pi_0}\hat{Q}_{\omega_1} = \hat{Q}_{\omega_1},$$
$$\implies \hat{Q}_{\omega_1} = Q^{\pi_0}(\cdot).$$

Our E-step amounts to calculating the Boltzmann distribution with $\varepsilon_{\omega_1} = 0$, which from Theorem 1 takes the form of a Dirac-delta distribution:

$$\pi_1(a|\cdot) = \delta\left(a = \arg\max_{a'} Q^{\pi_0}(a', \cdot)\right).$$

We can generalise to the $k$th EM update as:

$$\hat{Q}_{\omega_k}(\cdot) \leftarrow Q^{\pi_{k-1}}(\cdot), \tag{39}$$

$$\pi_k(a|\cdot) \leftarrow \delta\left(a = \arg\max_{a'} Q^{\pi_{k-1}}(a', \cdot)\right). \tag{40}$$

Together Eqs 39 and 40 are exactly the updates for policy iteration, an algorithm which is known to converge to the optimal policy [59, 55]. We therefore see that, even ignoring the constraint on $\Omega$, the optimal solution is still an attractive fixed point when our algorithms are carried out exactly. Using partial E- and M-steps give a variational approximation to the complete EM algorithm. We now provide a similar analysis using the fixed target of direct methods introduced in Appendix F.3.

**Analysis with Fixed Target**   Using a direct method, we replace the residual error with the fixed target residual error $\varepsilon_\omega \approx \varepsilon_{\omega,k} := \frac{c}{p}\|\mathcal{T}^{\pi_k}\hat{Q}_{\omega_k} - \hat{Q}_\omega\|_p^p$, giving the M-step update:

$$\omega_1 = \arg\min_\omega \varepsilon_{\omega,0}$$

which, under our assumption of representability, is achieved for

$$\varepsilon_{\omega_1,0} = 0,$$
$$\implies \hat{Q}_{\omega_1}(\cdot) = \mathcal{T}^{\pi_0}\hat{Q}_{\omega_0}.$$

As with our $\omega$-dependent target, the E-step amounts to calculating the Boltzmann distribution with $\varepsilon_{\omega_1,0} = 0$, which from Theorem 1 takes the form of a Dirac-delta distribution:

$$\pi_1(a|\cdot) = \delta\left(a = \arg\max_{a'} \hat{Q}_{\omega_1}(a', \cdot)\right).$$

We see that for any policy and function approximator, carrying out a complete E- and M- step results in a deterministic policy being learnt in this approximate regime. We generalise to the $k$th EM updates for $k > 2$ as:

$$\pi_{k-1}(a|\cdot) = \delta \left( a = \arg\max_{a'} \hat{Q}_{\omega_{k-1}}(a', \cdot) \right),$$

$$\omega_k = \arg\min_{\omega} \varepsilon_{\omega,k-1} = \arg\min_{\omega} \frac{c}{p} \|\mathcal{T}^{\pi_{k-1}} \hat{Q}_{\omega_{k-1}} - \hat{Q}_\omega\|_p^p,$$

$$\implies \varepsilon_{\omega_1,0} = 0,$$

$$\implies \hat{Q}_{\omega_k}(\cdot) = \mathcal{T}^{\pi_{k-1}} \hat{Q}_{\omega_{k-1}}(\cdot),$$

$$= r(\cdot) + \mathbb{E}_{s'|\cdot} \left[ \max_{a'} \hat{Q}_{\omega_{k-1}}(s', a') \right],$$

$$= \mathcal{T}^* \hat{Q}_{\omega_{k-1}}(\cdot). \tag{41}$$

From Eq. (41), we see that the EM algorithm with complete E- and M- steps implements $Q$-learning updates on our function approximator $\hat{Q}_{\omega_k}(\cdot) \leftarrow \mathcal{T}^* \hat{Q}_{\omega_{k-1}}(\cdot)$ for $k > 2$ [68]. See Yang et al. [72] for a theoretical exposition of the convergence this $Q$-learning algorithm using function approximators. When partial EM steps are carried out, we can view this algorithm as a variational approximation to $Q$-learning.

## G   Recovering MPO

We now derive the MPO objective from our framework. Under the probabilistic interpretation in Appendix B, the objective can be derived using the prior $p_\phi(h) = \mathcal{U}(s)\pi_\phi(a|s)$ instead of the uniform distribution. Following the same analysis as in Appendix B, this yields an action-posterior:

$$p_{\omega,\phi}(a|s, \mathcal{O}) = \frac{\exp\left(\frac{\hat{Q}_\omega(h)}{\varepsilon_\omega}\right) \pi_\phi(a|s)}{\int \exp\left(\frac{\hat{Q}_\omega(h)}{\varepsilon_\omega}\right) \pi_\phi(a|s) da}.$$

Again, following the same analysis as in Appendix B, our ELBO objective is:

$$\mathcal{L}(\omega, \theta, \phi) = \mathbb{E}_{d(s)} \left[ \mathbb{E}_{\pi_\theta(a|s)} \left[ \frac{\hat{Q}_\omega(h)}{\varepsilon_\omega} \right] - \mathrm{KL}(\pi_\theta(a|s) \| \pi_\phi(a|s)) \right]. \tag{42}$$

Including a hyper-prior $p(\phi)$ over $\phi$ adds an additional term to $\mathcal{L}(\omega, \theta, \phi)$:

$$\mathcal{L}(\omega, \theta, \phi) = \mathbb{E}_{d(s)} \left[ \mathbb{E}_{\pi_\theta(a|s)} \left[ \frac{\hat{Q}_\omega(h)}{\varepsilon_\omega} \right] - \mathrm{KL}(\pi_\theta(a|s) \| \pi_\phi(a|s)) \right] + \log p(\phi).$$

which is exactly the MPO objective, with an adaptive scaling constant $\varepsilon_\omega$ to balance the influence of $\mathrm{KL}(\pi_\theta(a|s) \| \pi_\phi(a|s))$. Without loss of generality, we ignore the hyperprior and analyse Eq. (42) instead.

As discussed by Abdolmaleki et al. [1], the MPO objective is similar to the PPO [52] objective with the KL-direction reversed. In our E-step, we find a new variational distribution $\pi_{\theta_{k+1}}(a|s)$ that maximises the ELBO with $\omega_k$ fixed: Doing so yields an identical E-step to MPO. In parametric form, we can use gradient ascent and apply the same analysis as in Appendix E.1, obtaining an update

**E-Step (MPO):**   $\theta_{i+1} \leftarrow \theta_i + \alpha_{\text{actor}} \left( \varepsilon_{\omega_k} \nabla_\theta \mathcal{L}(\omega_k, \phi_k, \theta)|_{\theta=\theta_i} \right),$

$$\varepsilon_{\omega_k} \nabla_\theta \mathcal{L}(\omega_k, \phi_k, \theta) = \mathbb{E}_{d(s)} \left[ \mathbb{E}_{\pi_\theta(a|s)} \left[ \hat{Q}_{\omega_k}(h) \nabla_\theta \log \pi_\theta(a|s) \right] - \varepsilon_{\omega_k} \nabla_\theta \mathrm{KL}(\pi_\theta(a|s) \| \pi_{\phi_k}(a|s)) \right]. \tag{43}$$

As a point of comparison, Abdolmaleki et al. [1] motivate the update in Eq. (43) by carrying out a partial E-step, maximising the "one-step" KL-regularised pseudo-likelihood objective. In our framework, maximising Eq. (43) constitutes a full E-step, without requiring approximation.

In our M-step, we maximise the LML using the posterior derived from the E-step, yielding the update:

**M-Step (MPO):** $\omega_{k+1}, \phi_{k+1} \leftarrow \arg\max_{\omega,\phi} \mathcal{L}(\omega, \phi, \theta_{k+1})$,

$$\arg\max_{\omega,\phi} \mathcal{L}(\omega, \phi, \theta_{k+1}) = \arg\max_{\omega,\phi} \left( \mathbb{E}_{d(s)} \left[ \mathbb{E}_{\pi_{\theta_{k+1}}} \left[ \frac{\hat{Q}_\omega(h)}{\varepsilon_\omega} \right] - \mathrm{KL}(\pi_{\theta_{k+1}} \parallel \pi_\phi(a|s)) \right] \right).$$

Maximising for $\phi$ can be achieved exactly by setting $\pi_\phi(a|s) = \pi_{\theta_{k+1}}(a|s)$, under which $\mathrm{KL}(\pi_{\theta_{k+1}} \parallel \pi_\phi(a|s)) = 0$. Maximising for $\omega$ is equivalent to finding $\arg\max_\omega \mathbb{E}_{d(s)\pi_{\theta_{k+1}}} \left[ \frac{\hat{Q}_\omega(h)}{\varepsilon_\omega} \right]$, which accounts for the missing policy evaluation step, and can be implemented using the gradient ascent updates from Eq. (7). Setting $\pi_\phi(a|s) = \pi_{\theta_{k+1}}(a|s)$ is exactly the M-step update for MPO and, like in TRPO [51], means that $\pi_\phi(a|s)$ can be interpreted as the old policy, which is updated only after policy improvement. The objective in Eq. (42) therefore prevents policy improvement from straying too far from the old policy, adding a penalisation term $\mathrm{KL}(\pi_\theta(a|s) \parallel \pi_{\mathrm{OLD}}(a|s))$ to the classic RL objective.

# H   Variational Actor-Critic Algorithm Pseudocode

Algorithms 1 and 2 show the pseudocode for the variational actor-critic algorithms *virel* and *beta* described in Section 5. The respective objectives are:

$$J^V(\phi) = \mathbb{E}_{s_t \sim \mathcal{D}} \left[ \frac{1}{2} \left( V_\phi(s_t) - \mathbb{E}_{a_t \sim \pi_\theta} [Q_\omega(s_t, a_t)] \right)^2 \right],$$

$$J^Q(\omega) = \mathbb{E}_{(h_t, r_t, s_{t+1}) \sim \mathcal{D}} \left[ \frac{1}{2} \left( r_t + \gamma V_{\bar{\phi}}(s_{t+1}) - Q_\omega(h_t) \right)^2 \right],$$

$$J^{\pi^q}_{virel}(\theta) = \mathbb{E}_{h_t \sim \mathcal{D}} \left[ \log \pi_\theta(a_t|s_t)(\alpha - (Q_\omega(h_t) - V_{\bar{\phi}}(s_t))) \right],$$

$$J^{\pi^q}_{beta}(\theta) = \mathbb{E}_{h_t \sim \mathcal{D}} \left[ \log \pi_\theta(a_t|s_t) \left( \frac{1-\gamma}{r_{avg}} \varepsilon_\omega - (Q_\omega(h_t) - V_{\bar{\phi}}(s_t)) \right) \right].$$

Note that the derivative of the policy objectives can be found using the reparametrisation trick [32, 29], which we use for our implementation.

---

| **Algorithm 1** Variational Actor-Critic: *virel* | **Algorithm 2** Variational Actor-Critic: *beta* |
|---|---|
| Initialize parameter vectors $\phi, \bar{\phi}, \theta, \omega, \mathcal{D} \leftarrow \{\}$ | Initialize parameter vectors $\phi, \bar{\phi}, \theta, \omega, \mathcal{D} \leftarrow \{\}$ |
| **for** each iteration **do** | **for** each iteration **do** |
|    **for** each environment step **do** |    **for** each environment step **do** |
|      $a_t \sim \pi^q(a|s;\theta)$ |      $a_t \sim \pi^q(a|s;\theta)$ |
|      $s_{t+1} \sim p(s_{t+1}|s_t, a_t)$ |      $s_{t+1} \sim p(s_{t+1}|s_t, a_t)$ |
|      $\mathcal{D} \leftarrow \mathcal{D} \cup \{(s_t, a_t, r(s_t, a_t), s_{t+1})\}$ |      $\mathcal{D} \leftarrow \mathcal{D} \cup \{(s_t, a_t, r(s_t, a_t), s_{t+1})\}$ |
|    **end for** |    **end for** |
|    **for** each gradient step **do** |    **for** each gradient step **do** |
|      $\phi \leftarrow \phi - \lambda_V \hat{\nabla}_\phi J^V(\phi)$ (M-step) |      $\varepsilon_\omega \leftarrow \mathbb{E}_\mathcal{D} \left[ \left( r_t + \gamma V_{\bar{\phi}}(s_{t+1}) - Q_\omega(h_t) \right)^2 \right]$ |
|      $\omega \leftarrow \omega - \lambda_Q \hat{\nabla}_\omega J^Q(\omega)$ (M-step) |      $\phi \leftarrow \phi - \lambda_V \hat{\nabla}_\phi J^V(\phi)$ (M-step) |
|      $\theta \leftarrow \theta - \lambda_{\pi^q} \hat{\nabla}_\theta J^{\pi^q}_{virel}(\theta)$ (E-step) |      $\omega \leftarrow \omega - \lambda_Q \hat{\nabla}_\omega J^Q(\omega)$ (M-step) |
|      $\bar{\phi} \leftarrow \tau\bar{\phi} + (1-\tau)\phi$ |      $\theta \leftarrow \theta - \lambda_{\pi^q} \hat{\nabla}_\theta J^{\pi^q}_{beta}(\theta)$ (E-step) |
|    **end for** |      $\bar{\phi} \leftarrow \tau\bar{\phi} + (1-\tau)\phi$ |
| **end for** |    **end for** |
| | **end for** |

# I   Experimental details

## I.1   Parameter Values

Note that instead of specifying temperature $c$, we fix $c = 1$ for all implementations and scale reward instead.

Table 1: Summary of Experimental Parameter Values

| PARAMETER | VALUE |
|---|---|
| Steps per evaluation | 1000 |
| Path Length | 999 |
| Discount factor | 0.99 |

**Mujoco-v2 Experiments:**

| | |
|---|---|
| Batch size | 128 |
| Net size | 300 |
| $\lambda_\beta \approx \dfrac{1-\gamma}{r_{avg}}$ | Humanoid<br>4e-4<br>All other<br>4e-3 |
| Reward scale | Hopper, Half-Cheetah<br>5<br>Walker<br>3<br>All other<br>1 |
| Value function learning rate | 3e-4 |
| Policy learning rate | 3e-4 |
| MLP layout as given in https://github.com/vitchyr/rlkit | |

**Mujoco-v1 Experiments:**

Values as used by Haarnoja et al. [25] in
https://github.com/haarnoja/sac

## I.2 Additional MuJoCo-v1 Experiments

Figure 7: Training curves on additional continuous control benchmarks Mujoco-v1.

## I.3 Additional MuJoCo-v2 Experiments

Figure 8: Training curves on additional continuous control benchmarks gym-Mujoco-v2.