[Reviews · NeurIPS 2019]

Reviewer 1



This paper brings an novel perspective on probabilistic frameworks for new reinforcement learning algorithms, and the adaptive temperature reweighting may lead to more insightful exploration built into our RL algorithms. The paper is written clearly, and is also well-organized and easy to understand, and the appendix is structured clearly as well, although the full length of the appendix + paper makes the paper a little unwieldy to read. The authors have clearly put in a lot of work into developing the theory and presentation in this paper, and although empirically the performance of the derived algorithms do not show significant improvement over max-ent RL methods (with twin Q functions as in TD3), the approach is interesting and I believe this paper would be well-suited for NeurIPS. Some specific comments: - In the definition of the residual error on L147, over what distribution is the L^p norm being referred to? - Instead of e_w being a global constant, have the authors considered parametrizing e_w as a function of h - this would allow for state-adaptive uncertainty and exploration, and I believe a majority of the results would still hold. - On L96, L227-229, the paper claims that MERLIN relies "on a variational distribution to approximate the underlying dynamics of the MDP for the entire trajectory". However, most works with the Max-Ent framework parametrize variational distributions through only the action distributions, and fix the variational distribution on dynamics to the actual dynamics model. The empirical evaluation on the Gym environments doesn't validate this hypothesis too strongly, but it would be interesting to see a more carefully designed test of this hypothesis. Perhaps results on environments that require long-horizon planning (where algorithms modelling full trajectories will be less performant) may be illuminating - Why were experiments run on different domains for the comparisons with the twin Q functions, than those run with a single Q function? - Putting an algorithm box or more extensive description of the evaluated algorithm in the main text would be useful, instead of just in the Appendix - How were hyperparameters chosen for all the algorithms?

Reviewer 2



Originality: The probabilistic model and variational inference approach is new and interesting. Related work is cited and it differs from previous work through a new adaptation of the temperature of the policies. Quality: I am not sure if I understand the definition of \epsilon_w in line 147. This seems to be recursive to me. Are you claiming that for any flexible Q there exist one (or many) softmax temperature \epsilon_w>0, so that the L^2 temporal difference error is exactly the softmax temperature? I get it in the limit but not for \epsilon_w>0 (proof?), also do you mean that the error is the same for any L^p norm? I have the same issue with the projected residual errors in Section F1. Further, the paper claims in line 231 "function approximator is used to model future dynamics, which is more expressive and better suited to capturing essential modes than a parametrised distribution". Can you explain why and give an example for this? Maybe related, the Boltzmann distribution might be more flexible than a simple tanh transformation of a Gaussian used in the Soft-Actor-Critic paper. Have you tried SAC with more flexible distributions having say multiple modes to see if the performance is due to the adaptation of the regularisation or just because of more flexible policies? Furthermore, as you consider some adaptation of the regulariser, can you compare your approach to entropy-regularised approaches that either reduce the entropy-penalty via some fixed learning rate (see for instance Geist et al, A theory of regularized markov decision processes, 2019) or optimized via gradient descent approaches (Haarnoja, et al, Soft actor-critic algorithms and applications, 2018)? Notwithstanding all these points, the submission seems technically sound in general with claims supported by theory (and with more technicalities than some related work) and experiments! Clarity: The paper is generally well written. I find the residual error in line 147 not so clear and find the introduction of the residual errors in Appendix F2 clearer and more plausible instead. Also there seems to be bit of a disconnect between the exposition of all the gradients in the main paper through a variational inference perspective, and then the algorithm pseudocode in the appendix that more or less uses policy improvement and Q-TD-error minimization. Can you elaborate more on those loss functions like J_virel(\theta), why do you have a constant \alpha there, does \epsilon_w depend on t in J_beta(\theta)? -Significance: The idea of a variational EM scheme to address RL is useful and I expect that others can build up on these ideas, either theoretically (like what can be said about performance errors) or empirically. The approach appears to be competitive with state-of-the art approaches. -Minor points: You say in line 969 that " minimising \eps{w,k} is the same as minimising the objective \eps_w from Section 3.1". Why? Is this not contradictory to lines 993-5? missing gradient after 843 w' versus \tilde{w} in 920 line 929 POST AUTHOR RESPONSE: I thank the authors for their feedback. Having read it along with the other reviews, I keep my initial score of 7. The rebuttal provides some clarification on the definition of \epsilon_w and indicates that for the Bellman operator, further theoretical work might be worthwile. They have also given some clarification concerning the flexibility of the parameterisation used for the policies. The authors also intend to reference additional related work that consider different types of adaptation for the entropy coefficient/penalty. While it would be nice to have some empirical comparison with such work, even without it, I think this is still a complete, long enough and interesting paper and I vote to accept it.

Reviewer 3



The article casts the control problem as a probabilistic inference one in a novel formulation based on a Boltzmann policy that uses the residual error as temperature. Since this policy has an intractable normalization constant variational inference is used through introducing another variational policy. The authors derive actor-critic algorithms through an expectation-maximization strategy applied to the variational objective. The authors offer extensive proofs for several useful properties of their objective: convergence under some assumptions, recovery of deterministic policies, etc/ The current work is also very well integrated with existing literature being motivated by the limitations of existing variational frameworks (the limited expressivity of the variational policy over trajectories and the difficulty to recover deterministic policies in maximum entropy approaches; and the risk-seeking policies pseudo-likelihood methods arrive at). The proposed method addresses all this limitations. Experiments with derived algorithms validate the approach by achieving state-of-the-art on a couple of continuous RL tasks. Baselines are relevant: a state-of-the-art algorithm (SAC), and another algorithm that naturally discovers deterministic policies (DDPG), the latter being closely related to one of the main claims in the article: that in the limit of the maximization of the objective the learned policy is deterministic. Considering the originality of the proposed objective, the strong theoretical treatment, the empirical validation, and also the nice exposition that places the article among related works, I propose for this paper to be accepted. Quality I consider the current article to be a high-quality presentation of a solid research effort. It does a good job in covering both theoretical and practical aspects (e.g. convergence proofs make some strong assumptions (2,4) that might be hard to meet in real setups, but discuss relaxations in the supplementary material). Originality The article builds on prior work as it starts with addressing some problems of the existing variational frameworks for RL, but it proposes an original Boltzmann policy that uses the residual error as an adaptive temperature. This strategy permits the derivation of a strategy for exploration until convergence based on the uncertainty in the optimality of the state-action value function. To the best of my knowledge, this is an original approach. Clarity The article is an excellent example of scientific writing. It does a good job in balancing the formal aspects (supported by detailed proofs in the supplementary material) with the intuition behind the different choices, and the connections with previous work (pseudo-likelihood and maximum entropy approaches). I think that in the current state one needs to have the supplementary material close in order to understand the proposed algorithms. I suggest moving into the article details such as the practical simplifications in appendix F3 (not with full detail, only enumerated in section 5). Section 5 mentions a claim from section 3.3 regarding soft value functions harming performance, but there is no such claim there. It is mentioned in section 2.2 though. Significance I consider the work to be important in the landscape of variational approaches to reinforcement learning as it solves known limitations of previous approaches and it’s both theoretically and empirically validated. Also, the empirical results show that algorithms might outperform existing algorithms on high dimensional inputs.

[Author Response · NeurIPS 2019]

We thank all reviewers for their insightful comments and the time they have spent carefully reviewing the paper.
Consistent among all reviewers is the comment that the paper could be improved with further experiments. In response
to this, we reiterate that our aim was to provide a novel framework with a theoretically sound interpretation of RL
as inference that simultaneously identifies and addresses the shortcomings of existing work while opening up new
classes of algorithms within this space that others can build upon. Our experiments were designed to provide empirical
evidence that our approach does not harm performance compared to state of the art, to support our theoretical claims
and demonstrate acceptable performance even when the most extreme approximations are used. While we feel the
submitted version already contains more than a conference paper's worth of material, we are already running some
additional experiments which, time and space permitting, we will include in the final version.

We will now address individual reviewer comments:

In response to Reviewer 1's third comment about modelling of entire trajectories in MERLIN, the algorithms for
MERLIN can all be obtained by considering the joint objective:

$$\mathcal{L}(\omega, \theta) := \mathbb{E}_{s \sim d(s)} \left[ \mathbb{E}_{a \sim \pi_\theta(a|s)} \left[ \frac{\hat{Q}_{\omega,soft}(h)}{\alpha} \right] \right],$$

where the variational distribution is $q_\theta(h) := d(s)\pi_\theta(a|s)$, the temperature constant is $\alpha$ and $\hat{Q}_{\omega,soft}(h)$ is the
parametrised approximation for the soft action value function. The above is equivalent to max-entropy formulation in
[Reinforcement Learning and Control as Probabilistic Inference: Tutorial and Review, Levine 2018] with a variational
approximation for the policy. However as the variational policy trains towards the Boltzmann distribution of the
soft action values with temperature $\alpha$, this indirectly takes into account the dependence of the entire trajectory on
the policy via $\hat{Q}_{\omega,soft}(h)$, and thus inadvertently ends up modelling entire trajectories despite grounding the MDP
dynamics.

In response to Reviewer 2's comment regarding the recursive definition of $\epsilon_\omega$, as we discuss in Appendix C, the
definition of $\varepsilon_\omega$ is recursive but only if using the simple Bellman operator for the Boltzmann policy. We introduce and
detail more complex operators in the set $\mathbb{T}$ that don't give a recursive definition in the Appendix. An example is the
optimal Bellman operator, which results in a Q-learning algorithm. In Appendix F.2, we introduce another operator that
recovers the optimal Bellman operator in a limit of sequences. Exploring further operators and investigating whether
for any flexible $Q$ there exist one (or many) consistent softmax temperatures $\varepsilon_\omega > 0$ when using the simple Bellman
operator for the Boltzmann policy is an exciting line of theoretical research for us, but one we feel is best saved for
future work.

In response to Reviewer 2's comment regarding comparisons to schemes where the adaptive entropy coefficient is
annealed according to a schedule or optimisation scheme, as we discuss in Appendix B, our formulation has a unique
Bayesian interpretation in that the entropy penalty is annealed according to the model uncertainty in the optimality of
$\hat{Q}_\omega(h)$. We thank the reviewer for drawing our attention to the references [A Theory of Regularized Markov Decision
Processes, Geist et al. 19] and [Soft Actor-Critic Algorithms and Applications, Haarnoja et al. 19], the former only
having been published since submitting to NeurIPS, and will extend the discussion accordingly.

Addressing Reviewer 2's second comment under the Quality section about function approximation for variational
policies, we implied that function approximation offers the choice to obtain arbitrary rich classes of variational
distributions that can, in principle, model the posterior conditional of action given the state exactly [Variational Inference
with Normalizing Flows, Rezende, 2015], instead of the simpler parametrisation involving Gaussian transformations.
We would like to clarify that the class of variational policies used in our experiments are the same for SAC and VIREL
(both using multi dimensional independent Gaussians), thus the experiments indeed demonstrate performance gains
from adaptive regularisation. We will clarify this difference in the paper.

Finally, we would like to thank Reviewer 3 for their careful analysis of the paper and will consider their sensible
suggestion of moving details from Appendix F3 into the main paper if the NeurIPS format permits. They are correct in
pointing out a reference typo in Section 5; we will update the paper accordingly.

[Meta-Review · NeurIPS 2019]

All reviewers were in agreement that the paper was interesting and well written.